# Curriculum Design for Teaching via Demonstrations: Theory and Applications

**Gaurav Yengera**[1,2]  **Rati Devidze**[1]  **Parameswaran Kamalaruban**[3]  **Adish Singla**[1]

gyengera@mpi-sws.org   rdevidze@mpi-sws.org   kparameswaran@turing.ac.uk   adishs@mpi-sws.org

[1]Max Planck Institute for Software Systems (MPI-SWS), Saarbrucken, Germany
[2]Saarland University, Saarland Informatics Campus (SIC), Saarbrucken, Germany
[3]The Alan Turing Institute, London, UK

## Abstract

We consider the problem of teaching via demonstrations in sequential decision-making settings. In particular, we study how to design a personalized curriculum over demonstrations to speed up the learner's convergence. We provide a unified curriculum strategy for two popular learner models: Maximum Causal Entropy Inverse Reinforcement Learning (MaxEnt-IRL) and Cross-Entropy Behavioral Cloning (CrossEnt-BC). Our unified strategy induces a ranking over demonstrations based on a notion of difficulty scores computed w.r.t. the teacher's optimal policy and the learner's current policy. Compared to the state of the art, our strategy doesn't require access to the learner's internal dynamics and still enjoys similar convergence guarantees under mild technical conditions. Furthermore, we adapt our curriculum strategy to the setting where no teacher agent is present using task-specific difficulty scores. Experiments on a synthetic car driving environment and navigation-based environments demonstrate the effectiveness of our curriculum strategy.

## 1 Introduction

Imitation learning is a paradigm in which a learner acquires a new set of skills by imitating a teacher's behavior. The importance of imitation learning is realized in real-world applications where the desired behavior cannot be explicitly defined but can be demonstrated easily. These applications include the settings involving both human-to-machine interaction [1–4], and human-to-human interaction [5, 6]. The two most popular approaches to imitation learning are Behavioral Cloning (BC) [7] and Inverse Reinforcement Learning (IRL) [8]. BC algorithms aim to directly match the behavior of the teacher using supervised learning methods. IRL algorithms operate in a two-step approach: first, a reward function explaining the teacher's behavior is inferred; then, the learner adopts a policy corresponding to the inferred reward.

In the literature, imitation learning has been extensively studied from the learner's point of view to design efficient learning algorithms [9–15]. However, much less work is done from the teacher's point of view to reduce the number of demonstrations required to achieve the learning objective. In this paper, we focus on the problem of Teaching via Demonstrations (TvD), where a helpful teacher assists the imitation learner in converging quickly by designing a personalized curriculum [16–20]. Despite a substantial amount of work on curriculum design for reinforcement learning agents [21–27], curriculum design for imitation learning agents is much less investigated.

Prior work on curriculum design for IRL learners has focused on two concrete settings: non-interactive and interactive. In the non-interactive setting [17, 18], the teacher provides a near-optimal set of demonstrations as a single batch. These curriculum strategies do not incorporate

any feedback from the learner, hence unable to adapt the teaching to the learner's progress. In the interactive setting [28], the teacher can leverage the learner's progress to adaptively choose the next demonstrations to accelerate the learning process. However, the existing state-of-the-art work [28] has proposed interactive curriculum algorithms that are based on learning dynamics of a specific IRL learner model (i.e., the learner's gradient update rule); see further discussion in Section 1.1. In contrast, we focus on designing an interactive curriculum algorithm with theoretical guarantees that is agnostic to the learner's dynamics. This will enable the algorithm to be applicable for a broad range of learner models, and in practical settings where the learner's internal model is unknown (such as tutoring systems with human learners). A detailed comparison between our curriculum algorithm and the prior state-of-the-art algorithms from [18, 28] is presented in Section 1.1.

Our approach is motivated by works on curriculum design for supervised learning and reinforcement learning algorithms that use a ranking over the training examples using a difficulty score [29–35]. In particular, our work is inspired by theoretical results on curriculum learning for linear regression models [32]. We define difficulty scores for any demonstration based on the teacher's optimal policy and the learner's current policy. We then study the differential effect of the difficulty scores on the learning progress for two popular imitation learners: Maximum Causal Entropy Inverse Reinforcement Learning (MaxEnt-IRL) [10] and Cross-Entropy loss-based Behavioral Cloning (CrossEnt-BC) [36]. Our main contributions are as follows:[1]

1. Our analysis for both MaxEnt-IRL and CrossEnt-BC learners leads to a unified curriculum strategy, i.e., a preference ranking over demonstrations. This ranking is obtained based on the ratio between the demonstration's likelihood under the teacher's optimal policy and the learner's current policy. Experiments on a synthetic car driving environment validate our curriculum strategy.

2. For the MaxEnt-IRL learner, we prove that our curriculum strategy achieves a linear convergence rate (under certain mild technical conditions), notably without requiring access to the learner's dynamics.

3. We adapt our curriculum strategy to the learner-centric setting where a teacher agent is not present through the use of task-specific difficulty scores. As a proof of concept, we show that our strategy accelerates the learning process in synthetic navigation-based environments.

## 1.1 Comparison to Existing Approaches on Curriculum Design for Imitation Learning

In the non-interactive setting, [18] have proposed a batch teaching algorithm (SCOT) by showing that the teaching problem can be formulated as a set cover problem. In contrast, our algorithm is interactive in nature and hence, can leverage the learner's progress (see experimental results in Section 5).

In the interactive setting, [28] have proposed the Omniscient algorithm (OMN) based on the iterative machine teaching (IMT) framework [37]. Their algorithm obtains strong convergence guarantees for the MaxEnt-IRL learner model; however, requires *exact* knowledge of the learner's dynamics (i.e, the learner's update rule). Our algorithm on the other hand is agnostic to the learner's dynamics and is applicable to a broader family of learner models (see Sections 4 and 5).

Also for the interactive setting, [28] have proposed the Blackbox algorithm (BBOX) as a heuristic to apply the OMN algorithm when the learner's dynamics are unknown—this makes the BBOX algorithm more widely applicable than OMN. However, this heuristic algorithm is still based on the gradient functional form of the linear MaxEnt-IRL learner model (see Footnote 2), and does not provide any convergence guarantees. In contrast, our algorithm is derived independent of any specific learner model and we provide a theoretical analysis of our algorithm for different learner models (see Theorems 1, 2, and 3). Another crucial difference is that the BBOX algorithm requires access to the true reward function of the environment, which precludes it from being applied to learner-centric settings where no teacher agent is present. In comparison, our algorithm is applicable to learner-centric settings (see experimental results in Section 6).

## 1.2 Additional Related Work on Curriculum Design and Teaching

**Curriculum design.** Curriculum design for supervised learning settings has been extensively studied in the literature. Early works present the idea of designing a curriculum comprising of tasks with increasing difficulty to train a machine learning model [29–31]. However, these approaches require

---

task-specific knowledge for designing heuristic difficulty measures. Recent works have tackled the problem of automating curriculum design [38, 39]. There is also an increasing interest in theoretically analyzing the impact of a curriculum (ordering) of training tasks on the convergence of supervised learner models [32, 40, 41]. In particular, our work builds on the idea of difficulty scores of the training examples studied in [32].

The existing results on curriculum design for sequential decision-making settings are mostly empirical in nature. Similar to the supervised learning settings, the focus on curriculum design for reinforcement learning settings has been shifted from hand-crafted approaches [34, 35] to automatic methods [21–23, 25]. We refer the reader to a recent survey [42] on curriculum design for reinforcement learning. The curriculum learning paradigm has also been studied in psychology literature [43–47]. One key aspect in these works has been to design algorithms that account for the pedagogical intentions of a teacher, which often aims to explicitly demonstrate specific skills rather than just provide an optimal demonstration for a task. We see our work as complementary to these.

**Machine teaching.** The algorithmic teaching problem considers the interaction between a teacher and a learner where the teacher's objective is to find an optimal training sequence to steer the learner towards a desired goal [37, 48–50]. Most of the work in machine teaching for supervised learning settings is on batch teaching where the teacher provides a batch of teaching examples at once without any adaptation. The question of how a teacher should adaptively select teaching examples for a learner has been addressed recently in supervised learning settings [51–56].

Furthermore, [16–18, 28, 57, 58] have studied algorithmic teaching for sequential decision-making tasks. In particular, [17, 18] have proposed batch teaching algorithms for an IRL agent, where the teacher decides the entire set of demonstrations to provide to the learner before any interaction. These teaching algorithms do not leverage any feedback from the learner. In contrast, as discussed in Section 1.1, [28] have proposed interactive teaching algorithms (namely OMN and BBOX) for an IRL agent, where the teacher takes into account how the learner progresses. The works of [57, 58] are complementary to ours and study algorithmic teaching when the learner has a different worldview than the teacher or has its own specific preferences.

## 2 Formal Problem Setup

Here, we formalize our problem setting which is based on prior work on sequential teaching [28, 37].

**Environment.** We consider an environment defined as a Markov Decision Process (MDP) $\mathcal{M} := \left( \mathcal{S}, \mathcal{A}, \mathcal{T}, \gamma, P_0, R^E \right)$, where the state and action spaces are denoted by $\mathcal{S}$ and $\mathcal{A}$, respectively. $\mathcal{T} : \mathcal{S} \times \mathcal{S} \times \mathcal{A} \to [0, 1]$ is the transition dynamics, $\gamma$ is the discounting factor, and $P_0 : \mathcal{S} \to [0, 1]$ is an initial distribution over states $\mathcal{S}$. A policy $\pi : \mathcal{S} \times \mathcal{A} \to [0, 1]$ is a mapping from a state to a probability distribution over actions. The underlying reward function is given by $R^E : \mathcal{S} \times \mathcal{A} \to \mathbb{R}$.

**Teacher-learner interaction.** We consider a setting with two agents: a teacher and a sequential learner. The teacher has access to the full MDP $\mathcal{M}$ and has a *target policy* $\pi^E$ (e.g., a near-optimal policy w.r.t. $R^E$). The learner knows the MDP $\mathcal{M}$ but not the reward function $R^E$, i.e., has only access to $\mathcal{M} \setminus R^E$. The teacher's goal is to provide an informative sequence of demonstrations to teach the policy $\pi^E$ to the learner. Here, a teacher's demonstration $\xi = \left\{ \left( s_\tau^\xi, a_\tau^\xi \right) \right\}_{\tau=0,1,\dots}$ is obtained by first choosing an initial state $s_0^\xi \in \mathcal{S}$ (where $P_0(s_0^\xi) > 0$) and then choosing a trajectory, sequence of state-action pairs, obtained by executing the policy $\pi^E$ in the MDP $\mathcal{M}$. The interaction between the teacher and the learner is formally described in Algorithm 1. For simplicity, we assume that the teacher directly observes the learner's policy $\pi_t^L$ at any time $t$. In practice, the teacher could approximately infer the policy $\pi_t^L$ by probing the learner and using Monte Carlo methods.

**Generic learner model.** Here, we describe a generic learner update rule for Algorithm 1. Let $\Theta \subseteq \mathbb{R}^d$ be a parameter space. The learner searches for a policy in the following parameterized policy space: $\Pi_\Theta := \{ \pi_\theta : \mathcal{S} \times \mathcal{A} \to [0, 1], \text{ where } \theta \in \Theta \}$. For the policy search, the learner sequentially minimizes a loss function $\ell$ that depends on the policy parameter $\theta$ and the demonstration $\xi = \left\{ \left( s_\tau^\xi, a_\tau^\xi \right) \right\}_\tau$ provided by the teacher. More concretely, we consider $\ell\left( \xi, \theta \right) := -\log \mathbb{P}\left( \xi | \theta \right)$, where $\mathbb{P}\left( \xi | \theta \right) = P_0(s_0^\xi) \cdot \prod_\tau \pi_\theta \left( a_\tau^\xi | s_\tau^\xi \right) \cdot \mathcal{T}\left( s_{\tau+1}^\xi | s_\tau^\xi, a_\tau^\xi \right)$ is the likelihood (probability) of the demon-

---

**Algorithm 1** Teacher-Learner Interaction

---

1: **Initialization:** Initial knowledge of learner $\pi_1^L$.
2: **for** $t = 1, 2, \ldots$ **do**
3:     Teacher observes the learner's current policy $\pi_t^L$.
4:     Teacher provides demonstration $\xi_t$ to the learner.
5:     Learner updates its policy to $\pi_{t+1}^L$ using $\xi_t$.

---

stration $\xi$ under policy $\pi_\theta$ in the MDP $\mathcal{M}$. At time $t$, upon receiving a demonstration $\xi_t$ provided by the teacher, the learner performs the following online projected gradient descent update: $\theta_{t+1} \leftarrow \text{Proj}_\Theta [\theta_t - \eta_t g_t]$, where $\eta_t$ is the learning rate, and $g_t = [\nabla_\theta \ell(\xi_t, \theta)]_{\theta=\theta_t}$. Note that the parameter $\theta_1$ reflects the initial knowledge of the learner. Given the learner's current parameter $\theta_t$ at time $t$, the learner's policy is defined as $\pi_t^L := \pi_{\theta_t}$.

**Teaching objective.** For any policy $\pi$, the value (total expected reward) of $\pi$ in the MDP $\mathcal{M}$ is defined as $V^\pi := \sum_{s,a} \sum_{\tau=0}^\infty \gamma^\tau \cdot \mathbb{P}\{S_\tau = s \mid \pi, \mathcal{M}\} \cdot \pi(a \mid s) \cdot R^E(s, a)$, where $\mathbb{P}\{S_\tau = s \mid \pi, \mathcal{M}\}$ denotes the probability of visiting the state $s$ after $\tau$ steps by following the policy $\pi$. Let $\pi^L$ denote the learner's final policy at the end of teaching. The performance of the policy $\pi^L$ (w.r.t. $\pi^E$) in $\mathcal{M}$ can be evaluated via $\left| V^{\pi^E} - V^{\pi^L} \right|$ [9, 59]. The teaching objective is to ensure that the learner's final policy $\epsilon$-*approximates* the teacher's policy, i.e., $\left| V^{\pi^E} - V^{\pi^L} \right| \leq \epsilon$. The teacher aims to provide an optimized sequence of demonstrations $\{\xi_t\}_{t=1,2,\ldots}$ to the learner to achieve the teaching objective. The teacher's performance is then measured by the number of demonstrations required to achieve this objective. Based on existing work [28, 37], we assume that $\exists \theta^* \in \Theta$ such that $\pi^E = \pi_{\theta^*}$ (we refer to $\theta^*$ as the *target teaching parameter*). Similar to [28], we assume that a smoothness condition holds in the policy parameter space: $|V^{\pi_\theta} - V^{\pi_{\theta'}}| \leq \mathcal{O}(f(\|\theta - \theta'\|)) \, \forall \theta, \theta' \in \Theta$. Then, the teaching objective in terms of $V^\pi$ convergence can be reduced to the convergence in the parameter space, i.e., we can focus on the quantity $\|\theta^* - \theta_t\|$.

## 3 Curriculum Design using Difficulty Scores

In this section, we introduce our curriculum strategy which is based on the concept of *difficulty scores* and is agnostic to the dynamics of the learner.

**Difficulty scores.** We begin by assigning a difficulty score $\Psi_\theta(\xi)$ for any demonstration $\xi$ w.r.t. a parameterized policy $\pi_\theta$ in the MDP $\mathcal{M}$. Inspired by difficulty scores for supervised learning algorithms [32], we consider a difficulty score which is directly proportional to the loss function $\ell$, i.e., $\Psi_\theta(\xi) \propto g(\ell(\xi, \theta))$, for a monotonically increasing function $g$. Setting $g(\cdot) = \exp(\cdot)$ leads to $\Psi_\theta(\xi) = \frac{1}{\prod_\tau \pi_\theta(a_\tau^\xi | s_\tau^\xi)}$ for MDPs with deterministic transition dynamics. Based on this insight, we define the following difficulty score which we use throughout our work.

**Definition 1.** *The difficulty score of a demonstration $\xi$ w.r.t. the policy $\pi_\theta$ in the MDP $\mathcal{M}$ is given by $\Psi_\theta(\xi) := \frac{1}{\prod_\tau \pi_\theta(a_\tau^\xi | s_\tau^\xi)}$.*

Intuitively, the difficulty score of a demonstration $\xi$ w.r.t. an agent's policy is inversely proportional to the preference of the agent to follow the demonstration. Demonstrations with a higher likelihood under the agent's policy (higher preference) have a lower difficulty score and vice versa. With the above definition, the difficulty scores for any demonstration $\xi$ w.r.t. the teacher's and learner's policies (at any time $t$) are respectively given by $\Psi^E(\xi) := \Psi_{\theta^*}(\xi)$ and $\Psi_t^L(\xi) := \Psi_{\theta_t}(\xi)$.

**General curriculum strategy.** Our curriculum strategy picks the next demonstration $\xi_t$ to provide to the learner based on a preference ranking induced by the teacher's and learner's difficulty scores. The difficulty score of a demonstration $\xi$ w.r.t. the teacher and learner (at any time t) is denoted by $\Psi^E$ and $\Psi_t^L$ respectively. Specifically, our curriculum strategy is given by:

$$\xi_t \leftarrow \arg\max_\xi \frac{\Psi_t^L(\xi)}{\Psi^E(\xi)}. \tag{1}$$

**Teacher-centric and learner-centric settings.** In the teacher-centric setting formalized in Section 2, our curriculum strategy utilizes the difficulty scores induced by the learner's current policy $\pi_t^L$ and the teacher's policy $\pi^E$. From Eq. (1) and Definition 1, we obtain the following teacher-centric curriculum strategy: $\xi_t \leftarrow \arg\max_\xi \prod_\tau \frac{\pi^E(a_\tau^\xi | s_\tau^\xi)}{\pi_t^L(a_\tau^\xi | s_\tau^\xi)}$.

Additionally, we also consider the learner-centric setting where a teacher agent is not present and the target policy $\pi^E$ is unknown. Here, the learner can benefit from designing a self-curriculum (i.e., automatically ordering demonstrations) based on its current policy $\pi_t^L$. We adapt our curriculum strategy to this setting by utilizing task-specific domain knowledge to define the teacher's difficulty score $\Psi^E(\xi)$ for any demonstration $\xi$. From Eq. (1), given the learner's current policy $\pi_t^L$ and the teacher's difficulty score $\Psi^E(\xi)$, the learner-centric curriculum strategy is given as follows: $\xi_t \leftarrow \arg\max_\xi \frac{1}{\Psi^E(\xi) \prod_\tau \pi_t^L(a_\tau^\xi | s_\tau^\xi)}$.

Note that our curriculum strategy only requires access to the learner's and teacher's policies ($\pi_t^L$ and $\pi^E$) and does not depend on the learner's internal dynamics (i.e, its update rule as mentioned in Section 2). This makes our approach more widely applicable to practical applications where it is often possible to infer an agent's policy, but the internal update rule is unknown.

## 4 Theoretical Analysis of Our Curriculum Strategy

In this section, we present the theoretical analysis of our curriculum strategy for two popular learner models, namely, MaxEnt-IRL and CrossEnt-BC. For our analysis, we consider the teacher-centric setting as introduced in Section 2. Our curriculum strategy obtains a preference ranking over the demonstrations to provide to the learner based on the difficulty scores (see Definition 1). To this end, we analyze the relationship between the difficulty scores (w.r.t. the teacher and the learner) of the provided demonstration and the teaching objective (convergence towards the target teaching parameter $\theta^*$) during each sequential update step of the learner.

Given two difficulty values $\psi^E, \psi^L \in \mathbb{R}$, we define the feasible set of demonstrations at time $t$ as $\mathcal{D}_t\left(\psi^E, \psi^L\right) := \left\{\xi : \Psi^E(\xi) = \psi^E \text{ and } \Psi_t^L(\xi) = \psi^L\right\}$. This set contains all demonstrations $\xi$ for which the teacher's difficulty score $\Psi^E(\xi)$ is equal to the value $\psi^E$, and the learner's difficulty score $\Psi^L(\xi)$ is equal to the value $\psi^L$. Let $\Delta_t\left(\psi^E, \psi^L\right)$ denote the expected convergence rate of the teaching objective at time $t$, given difficulty values $\psi^E$ and $\psi^L$:

$$\Delta_t\left(\psi^E, \psi^L\right) := \mathbb{E}_{\xi_t | \psi^E, \psi^L}[\|\theta^* - \theta_t\|^2 - \|\theta^* - \theta_{t+1}(\xi_t)\|^2], \tag{2}$$

where the expectation is w.r.t. the uniform distribution over the set $\mathcal{D}_t\left(\psi^E, \psi^L\right)$. Below, we analyse the differential effect of $\psi^E$ and $\psi^L$ on $\Delta_t\left(\psi^E, \psi^L\right)$, i.e., the effect of picking demonstrations with higher or lower difficulty scores on the learning progress.

### 4.1 Analysis for MaxEnt-IRL Learner

Here, we consider the popular MaxEnt-IRL learner model [10, 59, 60] in an MDP $\mathcal{M}$ with deterministic transition dynamics, i.e., $\mathcal{T} : \mathcal{S} \times \mathcal{S} \times \mathcal{A} \to \{0, 1\}$. The MaxEnt-IRL learner model uses a parametric reward function $R_\theta : \mathcal{S} \times \mathcal{A} \to \mathbb{R}$ where $\theta \in \mathbb{R}^d$ is a parameter. The reward function $R_\theta$ also depends on a feature mapping $\phi : \mathcal{S} \times \mathcal{A} \to \mathbb{R}^{d'}$ which encodes each state-action pair $(s, a)$ by a feature vector $\phi(s, a) \in \mathbb{R}^{d'}$. For our theoretical analysis, we consider $R_\theta$ with a linear form, i.e., $R_\theta(s, a) := \langle \theta, \phi(s, a) \rangle$ and $d = d'$. In our experiments, we go beyond these simplifications and consider environments with stochastic transition dynamics and non-linear reward functions.

Under the MaxEnt-IRL learner model, the parametric policy takes the following soft-Bellman form: $\pi_\theta(a|s) = \exp(Q_\theta(s, a) - V_\theta(s))$, where $V_\theta(s) = \log \sum_a \exp Q_\theta(s, a)$ and $Q_\theta(s, a) = R_\theta(s, a) + \gamma \sum_{s'} \mathcal{T}(s'|s, a) \cdot V_\theta(s')$. For any given $\theta$, the corresponding policy $\pi_\theta$ can be efficiently computed via the Soft-Value-Iteration procedure with reward $R_\theta$ (see [59, Algorithm. 9.1]). For the above setting and a given parameter $\theta$, the probability distribution $\mathbb{P}(\xi|\theta)$ over the demonstration $\xi$ takes the closed-form $\mathbb{P}(\xi|\theta) = \frac{\exp(\langle \theta, \mu^\xi \rangle)}{Z(\theta)}$, where $\mu^\xi := \sum_{\tau=0}^\infty \gamma^\tau \phi(s_\tau^\xi, a_\tau^\xi)$ and $Z(\theta)$ is a normalization factor. Then, at time $t$, the gradient of the MaxEnt-IRL learner is given by $g_t = \mu^{\pi_{\theta_t}} - \mu^{\xi_t}$, where

$\mu^\pi := \sum_{s,a} \sum_{\tau=0}^{\infty} \gamma^\tau \cdot \mathbb{P}\{S_\tau = s \mid \pi, \mathcal{M}\} \cdot \pi(a \mid s) \cdot \phi(s,a)$ is the feature expectation vector of policy $\pi$. We note that our curriculum strategy in Eq. (1) is not using knowledge of $g_t$.

For the MaxEnt-IRL learner, we obtain the following theorem, which shows the differential effect of the difficulty scores (w.r.t. the teacher and the learner) on the expected rate of convergence of the teaching objective $\Delta_t(\psi^E, \psi^L)$. We note that [32] obtained similar results for linear regression learner models in the supervised learning setting.

**Theorem 1.** *Assume that $\eta_t$ is sufficiently small for all $t$ s.t. $\eta_t \|g_t\|^2 \ll 2|\langle \theta^* - \theta_t, g_t \rangle|$, where $g_t$ is the gradient of the MaxEnt-IRL learner. Then, for the MaxEnt-IRL learner, the expected convergence rate of the teaching objective $\Delta_t(\psi^E, \psi^L)$ is:*

- *monotonically decreasing with value $\psi^E$, i.e., $\frac{\partial \Delta_t}{\partial \psi^E} < 0$, and*

- *monotonically increasing with value $\psi^L$, i.e., $\frac{\partial \Delta_t}{\partial \psi^L} > 0$.*

Theorem 1 suggests that choosing demonstrations with lower difficulty score w.r.t. the teacher's policy and higher difficulty score w.r.t. the learner's policy would lead to faster convergence. Our curriculum strategy in Eq. (1) induces a preference ranking over demonstrations that aligns with these insights of Theorem 1. Furthermore, the following theorem states that the particular form of combining the two difficulty scores used in curriculum strategy, Eq. (1), achieves linear convergence to the teaching objective. This is similar to the state-of-the-art OMN algorithm based on the IMT framework for sequential learners [28, 37]. Importantly, unlike the OMN algorithm, our curriculum strategy does not rely on specifics of the learner model when selecting demonstrations.

**Theorem 2.** *Consider Algorithm 1 with the MaxEnt-IRL learner and our curriculum strategy in Eq. (1). Then, the teaching objective $\|\theta^* - \theta_t\| \leq \epsilon$ is achieved in $t = \mathcal{O}(\log \frac{1}{\epsilon})$ iterations.*

In the above theorem, the constant terms suppressed by the $\mathcal{O}(\cdot)$ notation depend on the learning rate of the learner ($\eta_t$), the distance between the learner's initial parameter/knowledge and the target teaching parameter ($\|\theta^* - \theta_1\|$), and the *richness* of the set of demonstrations obtained by executing the policy $\pi^E$ in the MDP $\mathcal{M}$. The *richness* notion is formally discussed in the Appendix.

### 4.2 Analysis for CrossEnt-BC Learner

Next, we consider the CrossEnt-BC learner model [15, 36]. In this case, the learner's parametric policy takes the following softmax form: $\pi_\theta(a|s) = \frac{\exp(H_\theta(s,a))}{\sum_{a'} \exp(H_\theta(s,a'))}$, where $H_\theta : \mathcal{S} \times \mathcal{A} \to \mathbb{R}$ is a parametric scoring function that depends on the parameter $\theta \in \mathbb{R}^d$ and a feature mapping $\phi : \mathcal{S} \times \mathcal{A} \to \mathbb{R}^{d'}$. For our theoretical analysis, we consider a linear scoring function $H_\theta$ of the form $H_\theta(s,a) := \langle \theta, \phi(s,a) \rangle$ (with $d = d'$). Then, at time step $t$, the gradient $g_t$ of the CrossEnt-BC learner is given by: $g_t = \sum_{\tau=0}^{\infty} \left( \mathbb{E}_{a \sim \pi_{\theta_t}(\cdot | s_\tau^{\xi_t})} [\phi(s_\tau^{\xi_t}, a)] - \phi(s_\tau^{\xi_t}, a_\tau^{\xi_t}) \right)$. In the experiments, we also consider non-linear scoring functions parameterized by neural networks.

Similar to Theorem 1, we obtain the following theorem for the CrossEnt-BC learner, which also justifies our curriculum strategy in Eq. (1).

**Theorem 3.** *Assume that $\eta_t$ is sufficiently small for all $t$ s.t. $\eta_t \|g_t\|^2 \ll 2|\langle \theta^* - \theta_t, g_t \rangle|$, where $g_t$ is the gradient of the CrossEnt-BC learner. Then, for the CrossEnt-BC learner, the expected convergence rate of the teaching objective $\Delta_t(\psi^E, \psi^L)$, after first-order approximation, is:*

- *monotonically decreasing with $\psi^E$, i.e., $\frac{\partial \Delta_t}{\partial \psi^E} < 0$, and*

- *monotonically increasing with $\psi^L$, i.e., $\frac{\partial \Delta_t}{\partial \psi^L} > 0$.*

We note that the proof of Theorem 3 relies on the first-order Taylor approximation of the term $\sum_\tau \log \sum_{a'} \exp\left( H_\theta\left(s_\tau^{\xi_t}, a'\right) \right)$ around $\theta_t$ (detailed in the Appendix). Due to this approximation, it is more challenging to obtain a convergence result analogous to Theorem 2.

## 5 Experimental Evaluation: Teacher-Centric Setting

Inspired by the works of [28, 61, 62], we evaluate the performance of our curriculum strategy, Eq. (1), in a synthetic car driving environment on MaxEnt-IRL and CrossEnt-BC learners. In particular, we consider the environment of [28] and the teacher-centric setting of Section 2.

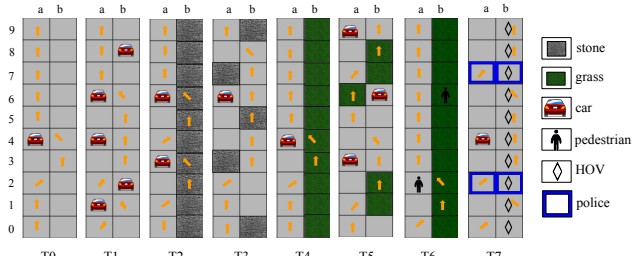

| $\phi^E(s)$ | $w$ |
|---|---|
| stone | -1 |
| grass | -0.5 |
| car | -5 |
| ped | -10 |
| car-front | -2 |
| ped-front | -5 |
| HOV | +1 |
| police | 0 |

Figure 1: Car environment with 8 different types of tasks. Arrows represent the path taken by the teacher's policy.

Figure 2: Environment features $\phi^E(s)$ and reward weights $w$.

**Car driving setup.** Fig. 1 illustrates a synthetic car driving environment consisting of 8 different types of tasks, denoted as T0, T1, ..., T7. Each type is associated with different driving skills. For instance, T0 corresponds to a basic setup representing a traffic-free highway. T1 represents a crowded highway. T2 has stones on the right lane, whereas T3 has a mix of both cars and stones. Similarly, T4 has grass on the right lane, and T5 has a mix of both grass and cars. T6 and T7 introduce more complex features such as pedestrians, police, and HOV (high occupancy vehicles). The agent starts navigating from an initial state at the bottom of the left lane of each task, and the goal is to reach the top of a lane while avoiding cars, stones, and other obstacles. The agent's action space is given by $\mathcal{A} = \{$left, straight, right$\}$. Action left steers the agent to the left of the current lane. If the agent is already in the leftmost lane when taking action left, then the lane is randomly chosen with uniform probability. We define similar stochastic dynamics for taking action right; action straight means no change in the lane. Irrespective of the action taken, the agent always moves forward.

**Environment MDP.** Based on the above setup, we define the environment MDP, $\mathcal{M}_{\text{car}}$, consisting of 8 types of tasks, namely T0–T7, and 5 tasks of each type. Every location in the environment is associated with a state. Each task is of length 10 and width 2, leading to a state space of size $5 \times 8 \times 20$. We consider an action-independent reward function $R^{E_{\text{car}}}$ that is dependent on an underlying feature vector $\phi^E$ (see Fig. 2). The feature vector of a state $s$, denoted by $\phi^E(s)$, is a binary vector encoding the presence or absence of an object at the state. In this work we have two types of features: features indicating the type of the current cell as stone, grass, car, ped, police, and HOV, as well as features providing some look-ahead information such as whether there is a car or pedestrian in the immediate front cell (denoted as car-front and ped-front). Now we explain the reward function $R^{E_{\text{car}}}$. For states in tasks of type T0-T6, the reward is given by $\langle w, \phi^E(s) \rangle$ (see Fig. 2). Essentially there are different penalties (i.e., negative rewards) for colliding with specific obstacles such as stone and car. For states in tasks of type T7, there is a reward of value +1 for driving on HOV; however, if police is present while driving on HOV, a reward value of $-5$ is obtained. Overall, this results in the reward function $R^{E_{\text{car}}}$ being nonlinear.

## 5.1 Teaching Algorithms

Here, we introduce the teaching algorithms considered in our experiments. The teacher's near-optimal policy $\pi^E$ is obtained via policy iteration [63]. The teacher selects demonstrations to provide to the learner using its teaching algorithm. We compare the performance of our proposed CUR teacher, which implements our strategy in Eq. (1), with the following baselines:

- CUR-T: A variant of our CUR teacher that samples demonstrations based on the difficulty score $\Psi^E$ alone, and sets $\Psi^L_t$ to constant.

- CUR-L: A similar variant of our CUR teacher that samples demonstrations based on the difficulty score $\Psi^L_t$ alone, and sets $\Psi^E$ to constant.

- AGN: an agnostic teacher that picks demonstrations based on random ordering [28, 32].

- OMN: The omniscient teacher is a state-of-the-art algorithm [28, 37], which is applicable only to MaxEnt-IRL learners. OMN requires complete knowledge of the parameter $\theta^*$, the learner's current parameter $\theta_t$, and the learner's gradients $\eta_t g_t$. Based on this knowledge, the teacher picks demonstrations to directly steer the learner towards $\theta^*$, i.e., by minimizing $\|\theta^* - (\theta_t - \eta_t g_t)\|$.

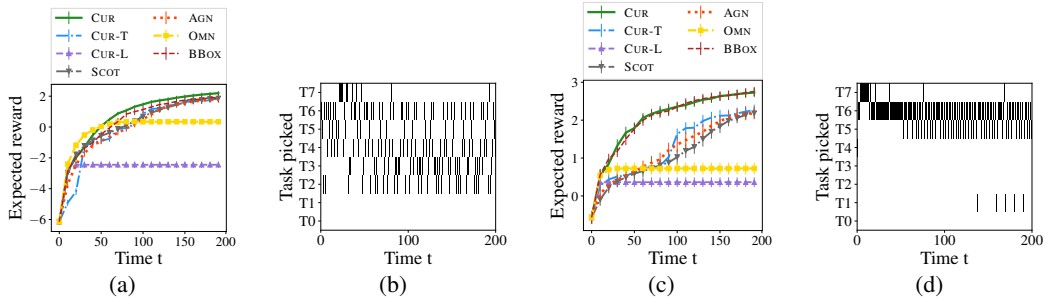

Figure 3: Learning curves and curriculum visualization for MaxEnt-IRL learners (with varying initial knowledge) trained on the car driving environment: (a) reward convergence plot and (b) curriculum generated by the CUR teacher for the learner with initial knowledge of T0; (c) reward convergence plot and (d) curriculum generated by the CUR teacher for the learner with initial knowledge of T0−T3.

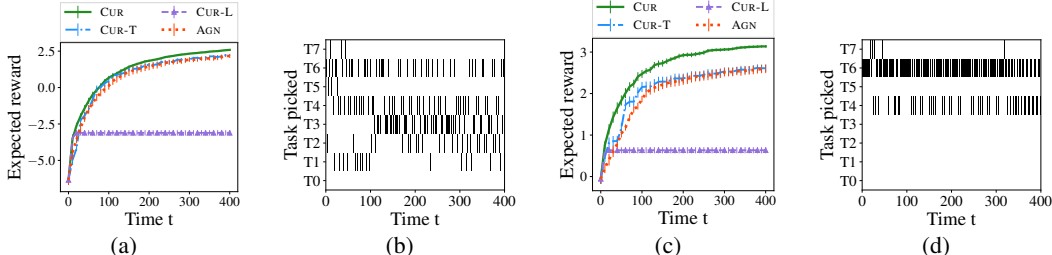

Figure 4: Learning curves and curriculum visualization for CrossEnt-BC learners (with varying initial knowledge) trained on the car driving environment: (a) reward convergence plot and (b) curriculum generated by the CUR teacher for the learner with initial knowledge of T0; (c) reward convergence plot and (d) curriculum generated by the CUR teacher for the learner with initial knowledge of T0−T3.

- BBOX: The blackbox teacher [28] is designed based on the functional form of gradients for the linear MaxEnt-IRL learner model.[2] Specifically, the teacher picks a demonstration $\xi$ which maximizes $|\sum_{s',a'}\{\rho^{\pi_t^L}(s',a') - \rho^\xi(s',a')\}R^E(s',a')|$, where $\rho$ denotes state visitation frequency vectors. The BBOX teacher does not require access to $\theta^*$ or the learner's current parameter $\theta_t$; however, it requires access to the true reward function $R^E$.

- SCOT: The SCOT teacher [18] aims to find the smallest set of demonstrations required to teach an optimal reward function to the MaxEnt-IRLlearner. The teacher uses a set cover algorithm to pre-compute the entire curriculum as a batch, prior to training. In our implementation, after having provided the entire batch, the teacher continues providing demonstrations selected at random.

### 5.2 Learner Models

Next, we describe the MaxEnt-IRL and CrossEnt-BC learner models. For the MaxEnt-IRL learner, we evaluate all the above-mentioned teaching algorithms that include state-of-the-art baselines; for the CrossEnt-BC learner, we evaluate CUR, CUR-T, CUR-L, and AGN algorithms.

**MaxEnt-IRL learner.** For alignment with the prior state-of-the-art work on teaching sequential MaxEnt-IRL learners [28], we perform *teaching over states* in our experiments. More concretely, at time $t$ the teacher picks a state $s_t$ (where $P_0(s_t) > 0$) and provides all demonstrations starting from $s_t$ to the learner given by $\Xi_{s_t} = \left\{\xi = \left\{\left(s_\tau^\xi, a_\tau^\xi\right)\right\}_\tau \text{ s.t. } s_0^\xi = s_t\right\}$. The gradient $g_t$ of the MaxEnt-IRL learner is then given by $g_t = \mu^{\pi_{\theta_t}, s_t} - \mu^{\Xi_{s_t}}$, where (i) $\mu^{\Xi_{s_t}} := \frac{1}{|\Xi_{s_t}|}\sum_{\xi \in \Xi_{s_t}} \mu^\xi$, and (ii) $\mu^{\pi, s_t}$ is the feature expectation vector of policy $\pi$ with starting state set to $s_t$ (see Section 4.1). Based on [28], we consider the learner's feature mapping as $\phi(s, a) = \phi^E(s)$ and the learner uses a non-linear parametric reward function $R_\theta^L(s, a) = \langle\theta_{1:d'}, \phi(s, a)\rangle + \langle\theta_{d'+1:2d'}, \phi(s, a)\rangle^2$ where $d'$ is the

---

[2]The BBOX teacher's objective is derived under the assumptions that the reward function can be linearly parameterized as $\langle w^*, \phi^E(s)\rangle$ and gradients $g_t$ are based on the linear MaxEnt-IRL learner model. Under these assumptions, the teacher's objective can be equivalently written as $|\langle w^*, g_t\rangle|$.

dimension of $\phi(s,a)$. As explained in [28], a linear reward representation cannot capture the optimal behaviour for $\mathcal{M}_{car}$. We consider learners with varying levels of initial knowledge, i.e., the learner is trained on a subset of tasks before the teaching process starts. In this setting, for our curriculum strategy in Eq. (1) the difficulty score of a set of demonstrations associated with a starting state $\Xi_s$ is computed as the mean difficulty score of individual demonstrations in the set.

**CrossEnt-BC learner.** We consider the CrossEnt-BC learner model of Section 4.2 as our second learner model. The learner's feature mapping is given by $\phi(s,a) = \mathbb{E}_{s' \sim \mathcal{T}(\cdot|s,a)}[\phi^E(s')]$. A quadratic parametric form is selected for the scoring function, i.e., $H_\theta(s,a) = \langle \theta_{1:d'}, \phi(s,a) \rangle + \langle \theta_{d'+1:2d'}, \phi(s,a) \rangle^2$, where $d'$ is the dimension of $\phi(s,a)$. We consider learners with varying initial knowledge and perform teaching over states similar to the MaxEnt-IRL learner.

## 5.3 Experimental results

Figs. 3a, 3c and 4a, 4c show the convergence of the total expected reward for the MaxEnt-IRL and CrossEnt-BC learners respectively, averaged over 10 runs. The CUR teacher outperforms OMN despite not requiring information about the learner's dynamics. For non-linear parametric reward functions, the MaxEnt-IRL learner no longer solves a convex optimization problem. As a result, forcing the learner to converge to a fixed parameter doesn't necessarily perform well, as seen by the poor performance of the OMN teacher in Fig. 3c. The CUR teacher is competitive with the BBOX teacher. Unlike our CUR teacher, the BBOX teacher does require exact access to the true reward function, $R^E$. The CUR teacher consistently outperforms the AGN and SCOT teachers, as well as both the CUR-T and CUR-L variants.

Figs. 3b, 3d and 4b, 4d visualize the curriculum generated by the CUR teacher for the MaxEnt-IRL and CrossEnt-BC learners respectively. Here, the curriculum refers to the type of task, T0−T7, associated with the demonstrations provided by the teacher to the learner at time step $t$. For both types of learners we see that at the beginning of training, the teacher focuses on tasks which teach skills the learner is yet to master. For example, in Fig. 4d, the teacher picks tasks T4, T6, and T7, which teaches the learner to avoid grass, pedestrians, and to navigate through police and HOV. We also notice that the CUR teacher can identify degradation in performance on previously mastered tasks, e.g., task T1 in Fig. 3d, and corrects for this by picking them again later during training.

**Additional results under limited observability.** In the above experiments, we consider the learner's policy to be fully observable by the teacher at every time step. Here, we study the performance of our CUR teacher under the limited observability setting, similar to [28], where the learner's policy needs to be estimated by probing the learner. The probing process is formally characterized by two parameters, $(B, k)$, where the learner's policy is probed after every $B$ time steps and each probing step corresponds to querying the learner's policy $\pi_t^L$ a total of $k$ times from each state $s \in \mathcal{S}$ in the MDP. The learner's policy, $\pi_t^L(a|s) \ \forall a, s$, is then approximated based on the fraction of the $k$ queries in which the learner performed action $a$ from state $s$. In between every $B$ time steps that the learner is probed, the CUR teacher does not update its estimate of the learner's policy. We

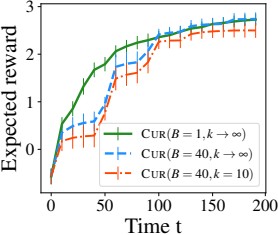

Figure 5: Learning curves for the MaxEnt-IRL learner under limited observability.

note that the $(B = 1, k \to \infty)$ setting corresponds to full observability of the learner. Fig. 5 depicts the performance of the CUR teacher for different values of $(B, k)$. Even under limited observability, the CUR teacher's performance is competitive with the full observability setting. The performance of $(B = 40, k \to \infty)$ is even slightly better at certain time steps during later stages of training compared to $(B = 1, k \to \infty)$, which is possibly due to the strategy of greedily picking demonstrations not being necessarily optimal. Also, for the limited observability setting it can be interesting to explore approaches that alleviate the need to query the full policy of the learner [64, 65].

## 6 Experimental Evaluation: Learner-Centric Setting

In this section, we evaluate our curriculum strategy in a learner-centric setting, i.e., no teacher agent is present, and the teacher's difficulty $\Psi^E(\xi)$ is expressed by a task-specific difficulty score (see Section 3). We evaluate our approach for training a multi-task neural policy to solve discrete optimization problems. Here, we provide an overview of the results with additional details in the Appendix.

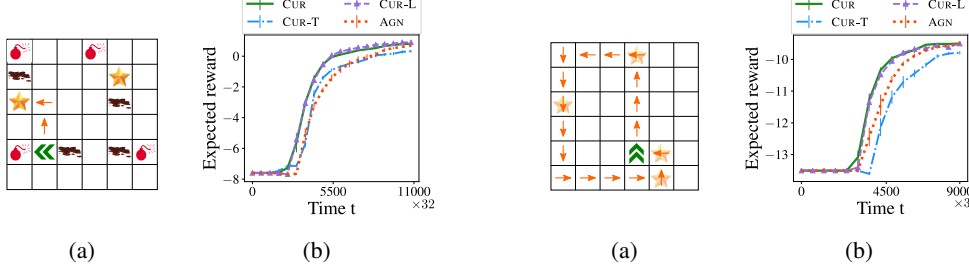

Figure 6: Illustration of a shortest path navigation task (left) and convergence curves (right).

Figure 7: Illustration of a travelling salesman navigation task (left) and convergence curves (right).

**Experiment setup.** We begin by describing the synthetic navigation-based environments considered in our experiments. Our first navigation environment comprises of tasks based on the shortest path problem [66]. We represent each task with a grid-world (see Fig. 6a) containing goal cells (depicted by stars), and cells with muds and bombs (shown in brown and red respectively). The agent aims to navigate to the closest goal cell while avoiding muds and bombs. Our second navigation environment comprises of tasks inspired by the travelling salesman problem (TSP) [67, 68] (see Fig. 7a). Again we represent each task with a grid-world, where the agent's goal is to find the shortest tour which visits all goals and returns to its initial location (see Fig. 7a). Orange arrows in Figs. 6a and 7a depict the optimal path for the agent. In our experimental setup, we begin by creating a pool of tasks and split them into training and test sets. The curriculum algorithms order the training tasks during the training phase based on their strategy to speed up the learner's progress. The aim of the learner is to learn a multi-task neural policy that can generalize to new unseen tasks in the test set.

**Curriculum algorithms.** We compare the performance of four different curriculum algorithms: (i) the CUR algorithm picks tasks from the training set using Eq. (1) where the numerator is $\Psi_t^L$ and the denominator $\Psi^E$ is defined by a task-specific difficulty score (detailed in the Appendix); (ii) the CUR-L algorithm picks tasks from the training set using Eq. (1) where the numerator is $\Psi_t^L$ and the denominator is set to 1; (iii) the CUR-T algorithm picks tasks from the training set using Eq. (1) where the numerator is set to 1 and the denominator is set to $\Psi^E$; (iv) the AGN algorithm picks tasks with a uniform distribution over the training set.

**Learner model.** We consider a neural CrossEnt-BC learner (see Section 4.2). The learner's scoring function $H_\theta$ is parameterized by a 6-layer Convolutional Neural Network (CNN). The CNN takes as input a feature mapping of the agent's current position in a task, and outputs a score for each action. The learner minimizes the cross-entropy loss between its predictions and the demonstrations.

**Results.** Figs. 6b and 7b, show the reward convergence curves on the test set for the different curriculum algorithms averaged over 5 runs. The CUR algorithm leads to faster reward convergence compared to the AGN algorithm, which is the common approach for training a neural policy. CUR-L is competitive with CUR in this setting which highlights the importance of the learner's difficulty.

## 7 Discussion and Conclusions

We presented a unified curriculum strategy, with theoretical guarantees, for the sequential MaxEnt-IRL and CrossEnt-BC learner models, based on the concept of difficulty scores. Our proposed strategy is independent of the learner's internal dynamics and is applicable in both teacher-centric and learner-centric settings. Experiments on a synthetic car driving environment and on navigation-based environments demonstrated the effectiveness of our curriculum strategy.

Our work provides theoretical underpinnings of curriculum design for teaching via demonstrations, which can be beneficial in educational applications such as tutoring systems and also for self-curriculum design for imitation learners. As such we do not see any negative societal impact of our work. Some of the interesting directions for future work include: obtaining convergence bounds for CrossEnt-BC and other learner models, designing curriculum algorithms for reinforcement learning agents based on the concept of difficulty scores, and designing approaches to efficiently approximate the learner's policy using less queries.

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
