# A  List of Appendices

In this section, we provide a brief description of the content in the appendices of the paper.

- Appendix B provides proofs for MaxEnt-IRL learner.
- Appendix C provides proofs for CrossEnt-BC learner.
- Appendix D provides a detailed description of the synthetic navigation-based experiments.

# B  Proofs for MaxEnt-IRL Learner

## B.1  Auxiliary Lemma

**Lemma 1.** *Consider the MaxEnt-IRL learner defined in Section 4.1. Then, at time step $t$, we have*

$$- \langle \theta^* - \theta_t, g_t \rangle \;=\; \log \frac{\Psi_t^L (\xi_t)}{\Psi^E (\xi_t)} + K_t,$$

*where $K_t = \log \frac{Z(\theta^*)}{Z(\theta_t)} - \langle \theta^* - \theta_t, \mu^{\pi_{\theta_t}} \rangle$ is a constant independent of $\xi_t$.*

*Proof.* Consider the following:

$$
\begin{aligned}
\Psi_\theta (\xi_t) \;&=\; \frac{1}{\prod_\tau \pi_\theta \left( a_\tau^{\xi_t} | s_\tau^{\xi_t} \right)} \\
&=\; \frac{P_0(s_0^{\xi_t})}{P_0(s_0^{\xi_t}) \cdot \prod_\tau \pi_\theta \left( a_\tau^{\xi_t} | s_\tau^{\xi_t} \right)} \\
&\overset{(i)}{=}\; \frac{P_0(s_0^{\xi_t})}{P_0(s_0^{\xi_t}) \cdot \prod_\tau \pi_\theta \left( a_\tau^{\xi_t} | s_\tau^{\xi_t} \right) \cdot \mathcal{T} \left( s_{\tau+1}^{\xi_t} | s_\tau^{\xi_t}, a_\tau^{\xi_t} \right)} \\
&=\; \frac{P_0(s_0^{\xi_t})}{\mathbb{P} \left( \xi_t | \theta \right)} \\
&=\; \frac{P_0(s_0^{\xi_t}) \cdot Z(\theta)}{\exp \left( \langle \theta, \mu^{\xi_t} \rangle \right)}
\end{aligned}
$$

where (i) is due to the deterministic transition dynamics of the MDP. Thus, we have:

$$\langle \theta, \mu^{\xi_t} \rangle \;=\; \log \frac{P_0(s_0^{\xi_t}) \cdot Z(\theta)}{\Psi_\theta (\xi_t)}.$$

Then, we get:

$$
\begin{aligned}
\left\langle \theta^* - \theta_t, \mu^{\xi_t} - \mu^{\pi_{\theta_t}} \right\rangle \;&=\; \log \frac{P_0(s_0^{\xi_t}) \cdot Z(\theta^*)}{\Psi_{\theta^*} (\xi_t)} - \log \frac{P_0(s_0^{\xi_t}) \cdot Z(\theta_t)}{\Psi_{\theta_t} (\xi_t)} - \langle \theta^* - \theta_t, \mu^{\pi_{\theta_t}} \rangle \\
&=\; \log \frac{\Psi_{\theta_t} (\xi_t)}{\Psi_{\theta^*} (\xi_t)} \cdot \frac{Z(\theta^*)}{Z(\theta_t)} - \langle \theta^* - \theta_t, \mu^{\pi_{\theta_t}} \rangle \\
&=\; \log \frac{\Psi_t^L (\xi_t)}{\Psi^E (\xi_t)} + K_t,
\end{aligned}
$$

where $K_t = \log \frac{Z(\theta^*)}{Z(\theta_t)} - \langle \theta^* - \theta_t, \mu^{\pi_{\theta_t}} \rangle$ is a constant independent of $\xi_t$. $\qquad\square$

## B.2  Proof of Theorem 1

**Technical conditions.** Let $\Theta = \mathbb{R}^d$, and for each time step $t$, the learning rate $\eta_t$ satisfies the following condition:

$$\eta_t^2 \, \|g_t\|^2 \;\ll\; 2\eta_t \, |\langle \theta^* - \theta_t, g_t \rangle| , \tag{3}$$

where $g_t$ is given in Section 4.1. We decompose the gradient as $g_t = \kappa \left(\theta^* - \theta_t\right) + \delta$, where $\delta \perp \left(\theta^* - \theta_t\right)$, and $\kappa \in \mathbb{R}$. Then, the above condition can be reduced to the following:

$$\eta_t^2 \left(|\kappa|^2 \left\|\theta^* - \theta_t\right\|^2 + \|\delta\|^2\right) \ll 2\eta_t |\kappa| \left\|\theta^* - \theta_t\right\|^2$$

$$\implies \quad \eta_t \ll \frac{2 |\kappa| \left\|\theta^* - \theta_t\right\|^2}{|\kappa|^2 \left\|\theta^* - \theta_t\right\|^2 + \|\delta\|^2}.$$

When the gradient $g_t$ primarily aligns with $\pm \left(\theta^* - \theta_t\right)$, and has a small magnitude to control variance, the above condition further simplifies as follows:

$$\eta_t \ll \frac{2}{|\kappa|}.$$

Smaller values of $|\kappa|$ would impose less stringent condition on $\eta_t$. From Lemma 1, one can easily observe that our curriculum strategy indeed aims to align the gradient $g_t$ with $-\left(\theta^* - \theta_t\right)$:

$$\arg\max_\xi \frac{\Psi_t^L (\xi)}{\Psi^E (\xi)} = \arg\max_\xi \log \frac{\Psi_t^L (\xi)}{\Psi^E (\xi)} = \arg\max_\xi \left\{- \left\langle \theta^* - \theta_t, g_t (\xi)\right\rangle\right\}.$$

We remark that these technical conditions are only required for our theoretical analysis, and not for our experiments.

*Proof.* Consider the following:

$$\begin{aligned}
\Delta_t \left(\psi^E, \psi^L\right) &= \mathbb{E}_{\xi_t | \psi^E, \psi^L} \left[\left\|\theta^* - \theta_t\right\|^2 - \left\|\theta^* - \theta_{t+1} (\xi_t)\right\|^2\right] \\
&= \mathbb{E}_{\xi_t | \psi^E, \psi^L} \left[\left\|\theta^* - \theta_t\right\|^2 - \left\|\theta^* - \theta_t + \eta_t g_t\right\|^2\right] \\
&= \mathbb{E}_{\xi_t | \psi^E, \psi^L} \left[-\eta_t^2 \left\|g_t\right\|^2 - 2\eta_t \left\langle\theta^* - \theta_t, g_t\right\rangle\right] \\
&\overset{(i)}{\approx} 2\eta_t \mathbb{E}_{\xi_t | \psi^E, \psi^L} \left[- \left\langle\theta^* - \theta_t, g_t\right\rangle\right] \\
&\overset{(ii)}{=} 2\eta_t \mathbb{E}_{\xi_t | \psi^E, \psi^L} \left[\log \frac{\Psi_t^L (\xi_t)}{\Psi^E (\xi_t)} + K_t\right] \\
&= 2\eta_t \log \frac{\psi^L}{\psi^E} + 2\eta_t K_t, \quad (4)
\end{aligned}$$

where the approximation (i) is due to Eq. (3), and (ii) is due to Lemma 1. Then, from (4), we have:

$$\frac{\partial \Delta_t}{\partial \psi^E} \approx -\frac{2\eta_t}{\psi^E} < 0, \text{ and}$$

$$\frac{\partial \Delta_t}{\partial \psi^L} \approx \frac{2\eta_t}{\psi^L} > 0.$$

$\square$

## B.3 Proof of Theorem 2

*Proof.* From Lemma 1, we have that

$$\arg\max_\xi \left\langle\theta^* - \theta_t, \mu^\xi\right\rangle = \arg\max_\xi \log \frac{\Psi_t^L (\xi)}{\Psi^E (\xi)} = \arg\max_\xi \frac{\Psi_t^L (\xi)}{\Psi^E (\xi)}.$$

Thus, our curriculum teaching algorithm picks the demonstration to provide by optimizing the following objective:

$$\xi_t \leftarrow \arg\max_\xi \left\langle\theta^* - \theta_t, \mu^\xi\right\rangle.$$

For a bounded feature mapping $\phi$, we have that $\left\|\mu^\xi\right\| \leq L, \forall \xi$. Any optimal solution $\xi_t$ to the above problem satisfies: $\mu^{\xi_t} = \frac{L}{\|\theta^* - \theta_t\|} \left(\theta^* - \theta_t\right)$. Since in our setting the teacher's demonstrations are

restricted to trajectories obtained by executing policy $\pi^E$ in the MDP $\mathcal{M}$, we assume that within the set of available teacher's demonstrations, the optimal feature vector has the following form [28, 37]:

$$\mu^{\xi_t} = \beta_t \left(\theta^* - \theta_t\right) + \delta_t,$$

where $\beta_t \in \left[0, \frac{L}{\|\theta^* - \theta_t\|}\right]$ bounds the magnitude of the gradient in the desired direction of $(\theta^* - \theta_t)$, and $\delta_t$ represents the deviation from the desired direction, s.t. $\Delta = \max_t \|\delta_t\|$. We further define the following terms: $z_{\max} = \max_t \|\theta^* - \theta_t\|$, $\eta_{\max} = \max_t \eta_t$, and $\beta = \min_t \eta_t \beta_t$.

Consider the following:

$$
\begin{aligned}
\|\theta^* - \theta_{t+1}\|^2 &= \left\|\theta^* - \left(\theta_t + \eta_t \mu^{\xi_t} - \eta_t \mu^{\pi_t^L}\right)\right\|^2 \\
&= \|\theta^* - \theta_t\|^2 + \eta_t^2 \left\|\mu^{\xi_t} - \mu^{\pi_t^L}\right\|^2 - 2\eta_t \left\langle \theta^* - \theta_t, \mu^{\xi_t} - \mu^{\pi_t^L}\right\rangle \\
&= \|\theta^* - \theta_t\|^2 + \eta_t^2 \left\|\beta_t (\theta^* - \theta_t) + \delta_t - \mu^{\pi_t^L}\right\|^2 - 2\eta_t \left\langle \theta^* - \theta_t, \beta_t (\theta^* - \theta_t) + \delta_t - \mu^{\pi_t^L}\right\rangle \\
&= \|\theta^* - \theta_t\|^2 + \eta_t^2 \beta_t^2 \|\theta^* - \theta_t\|^2 + \eta_t^2 \|\delta_t\|^2 + \eta_t^2 \left\|\mu^{\pi_t^L}\right\|^2 + 2\eta_t^2 \beta_t \langle \theta^* - \theta_t, \delta_t \rangle - 2\eta_t^2 \beta_t \left\langle \theta^* - \theta_t, \mu^{\pi_t^L}\right\rangle \\
&\quad - 2\eta_t^2 \left\langle \delta_t, \mu^{\pi_t^L}\right\rangle - 2\eta_t \beta_t \|\theta^* - \theta_t\|^2 - 2\eta_t \langle \theta^* - \theta_t, \delta_t \rangle + 2\eta_t \left\langle \theta^* - \theta_t, \mu^{\pi_t^L}\right\rangle \\
&\overset{(i)}{\leq} \left(1 + \eta_t^2 \beta_t^2 - 2\eta_t \beta_t\right) \|\theta^* - \theta_t\|^2 + \eta_t^2 \left[\Delta^2 + L^2\right] \\
&\quad + 2\eta_t \left(1 - \eta_t \beta_t\right) \Delta \|\theta^* - \theta_t\| + 2\eta_t \left(1 - \eta_t \beta_t\right) L \|\theta^* - \theta_t\| + 2\eta_t^2 \Delta L \\
&\overset{(ii)}{\leq} \left(1 - \eta_t \beta_t\right)^2 \|\theta^* - \theta_t\|^2 + \eta_t^2 \left(\Delta + L\right)^2 + 2\eta_t \left(1 - \eta_t \beta_t\right) \left(\Delta + L\right) z_{\max} \\
&\overset{(iii)}{\leq} \left(1 - \beta\right)^2 \|\theta^* - \theta_t\|^2 + \eta_{\max}^2 \left(\Delta + L\right)^2 + 2\eta_{\max} \left(1 - \beta\right) \left(\Delta + L\right) z_{\max} \\
&\overset{(iv)}{\leq} \left(1 - \beta\right)^2 \|\theta^* - \theta_t\|^2 + \eta_{\max} \left\{1 + 2 \left(1 - \beta\right) z_{\max}\right\} \left(\Delta + L\right),
\end{aligned}
$$

where (i) uses the inequalities $\left\|\mu^{\xi_t}\right\| \leq L$, and $\|\delta_t\| \leq \Delta$, along with the Cauchy-Schwarz inequality; (ii) utilizes the fact that $\|\theta^* - \theta_t\| \leq z_{\max}$; (iii) is obtained by substituting $\beta = \min_t \eta_t \beta_t$, and $\eta_{\max} = \max_t \eta_t$; (iv) is obtained when $\eta_{\max} \left(\Delta + L\right) \leq 1$. Note that the inequality (i) is valid when $1 - \eta_t \beta_t > 0, \forall t$.

With the inequality $\sqrt{a + b} \leq \sqrt{a} + \sqrt{b}$ for positive $a, b$, and utilizing recurrence, we obtain:

$$
\begin{aligned}
\|\theta^* - \theta_{t+1}\| &\leq \left(1 - \beta\right) \|\theta^* - \theta_t\| + \sqrt{\eta_{\max} \left\{1 + 2 \left(1 - \beta\right) z_{\max}\right\} \left(\Delta + L\right)} \\
&\leq \left(1 - \beta\right)^t \|\theta^* - \theta_1\| + \sqrt{\eta_{\max} \left\{1 + 2 \left(1 - \beta\right) z_{\max}\right\} \left(\Delta + L\right)} \sum_{s=0}^{\infty} \left(1 - \beta\right)^s \\
&= \left(1 - \beta\right)^t \|\theta^* - \theta_1\| + \sqrt{\eta_{\max} \left\{1 + 2 \left(1 - \beta\right) z_{\max}\right\} \left(\Delta + L\right)} \cdot \frac{1}{\beta} \\
&\leq \frac{\epsilon}{2} + \frac{\epsilon}{2} = \epsilon,
\end{aligned}
$$

for $t = \left(\log \frac{1}{1 - \beta}\right)^{-1} \log \frac{2\|\theta^* - \theta_1\|}{\epsilon} = \mathcal{O}\left(\log \frac{1}{\epsilon}\right)$, and $\eta_{\max} \left(\Delta + L\right) \leq \frac{\epsilon^2 \beta^2}{4\{1 + 2(1 - \beta)z_{\max}\}}$. $\qquad \square$

## C  Proofs for CrossEnt-BC Learner

### C.1  Auxiliary Lemma

**Lemma 2.** *Consider the CrossEnt-BC learner defined in Section 4.2. Then, at time step t, we have*

$$-\langle \theta^* - \theta_t, g_t \rangle \approx \log \frac{\Psi_t^L \left(\xi_t\right)}{\Psi^E \left(\xi_t\right)}.$$

*Proof.* Consider the following:

$$
\begin{aligned}
\log \Psi_\theta\left(\xi_t\right) &= -\log \prod_\tau \pi_\theta\left(a_\tau^{\xi_t} \mid s_\tau^{\xi_t}\right) \\
&= -\sum_\tau \log \pi_\theta\left(a_\tau^{\xi_t} \mid s_\tau^{\xi_t}\right) \\
&= \sum_\tau \log \sum_{a'} \exp\left(H_\theta\left(s_\tau^{\xi_t}, a'\right)\right) - \sum_\tau H_\theta\left(s_\tau^{\xi_t}, a_\tau^{\xi_t}\right) \\
&\overset{(i)}{\approx} \sum_\tau \log \sum_{a'} \exp\left(H_{\theta_t}\left(s_\tau^{\xi_t}, a'\right)\right) - \left\langle \theta - \theta_t, \sum_\tau \mathbb{E}_{a' \sim \pi_{\theta_t}\left(\cdot \mid s_\tau^{\xi_t}\right)}\left[\phi(s_\tau^{\xi_t}, a')\right] \right\rangle - \left\langle \theta, \sum_\tau \phi(s_\tau^{\xi_t}, a_\tau^{\xi_t}) \right\rangle
\end{aligned}
$$

where (i) is due to the first-order Taylor approximation of $\sum_\tau \log \sum_{a'} \exp\left(H_\theta\left(s_\tau^{\xi_t}, a'\right)\right)$ around $\theta_t$. Then, we have:

$$
\begin{aligned}
\log \frac{\Psi_{\theta_t}\left(\xi_t\right)}{\Psi_{\theta^*}\left(\xi_t\right)} &= \log \Psi_{\theta_t}\left(\xi_t\right) - \log \Psi_{\theta^*}\left(\xi_t\right) \\
&\approx \left\langle \theta^* - \theta_t, \sum_\tau \phi(s_\tau^{\xi_t}, a_\tau^{\xi_t}) - \sum_\tau \mathbb{E}_{a' \sim \pi_{\theta_t}\left(\cdot \mid s_\tau^{\xi_t}\right)}\left[\phi(s_\tau^{\xi_t}, a')\right] \right\rangle \\
&= -\left\langle \theta^* - \theta_t, g_t \right\rangle
\end{aligned}
$$

$\square$

## C.2 Proof of Theorem 3

**Technical conditions.** Let $\Theta = \mathbb{R}^d$, and for each time step $t$, the learning rate $\eta_t$ satisfies the following condition:

$$
\eta_t^2 \|g_t\|^2 \ll 2\eta_t \left|\langle \theta^* - \theta_t, g_t \rangle\right|, \tag{5}
$$

where $g_t$ is the gradient of the CrossEnt-BC learner as given in section 4.2. We can further simplify the above condition, similar to Section B.2.

*Proof.* Consider the following:

$$
\begin{aligned}
\Delta_t\left(\psi^E, \psi^L\right) &= \mathbb{E}_{\xi_t \mid \psi^E, \psi^L}\left[\|\theta^* - \theta_t\|^2 - \|\theta^* - \theta_{t+1}\left(\xi_t\right)\|^2\right] \\
&= \mathbb{E}_{\xi_t \mid \psi^E, \psi^L}\left[\|\theta^* - \theta_t\|^2 - \|\theta^* - \theta_t + \eta_t g_t\|^2\right] \\
&= \mathbb{E}_{\xi_t \mid \psi^E, \psi^L}\left[-\eta_t^2 \|g_t\|^2 - 2\eta_t \langle \theta^* - \theta_t, g_t \rangle\right] \\
&\overset{(i)}{\approx} 2\eta_t \mathbb{E}_{\xi_t \mid \psi^E, \psi^L}\left[-\langle \theta^* - \theta_t, g_t \rangle\right] \\
&\overset{(ii)}{\approx} 2\eta_t \mathbb{E}_{\xi_t \mid \psi^E, \psi^L}\left[\log \frac{\Psi_t^L\left(\xi_t\right)}{\Psi^E\left(\xi_t\right)}\right] \\
&= 2\eta_t \log \frac{\psi^L}{\psi^E}, \tag{6}
\end{aligned}
$$

where the approximation (i) is due to Eq. (5), and (ii) is due to Lemma 2. Then, from (6), we have:

$$
\frac{\partial \Delta_t}{\partial \psi^E} \approx -\frac{2\eta_t}{\psi^E} < 0, \text{ and}
$$

$$
\frac{\partial \Delta_t}{\partial \psi^L} \approx \frac{2\eta_t}{\psi^L} > 0.
$$

$\square$

## D Additional Details for Learner-Centric Experiments

In this appendix, we present additional experimental details for the synthetic navigation-based environments considered in Section 6.

### D.1 Environment MDPs

We first formally define the environment MDPs for the shortest path and TSP inspired environments described in Section 6.

**Shortest path environment.** A task in the shortest path environment is represented by a grid-world containing the agent, goals, muds, and bombs. Each possible configuration of a grid-world, including the agent's location and direction, is associated with a state in the shortest path environment MDP, $\mathcal{M}_{\text{path}}$. The size of the state space sees a combinatorial growth with the size of the grid, corresponding to different ways of placing bombs/muds/goals. Hence, the state-space is intractably large to enumerate. The agent's action space consists of 3 actions, $\mathcal{A} = \{\texttt{move}, \texttt{left}, \texttt{right}\}$. The actions $\texttt{left}$ or $\texttt{right}$ changes the agent's direction accordingly. The agent moves one step forward in its current direction with the action $\texttt{move}$. The environment reward function $R^{E_{\text{path}}}$ has a $-1$ reward value for each action performed by the agent. Reaching a goal cell has a $+10$ reward value. There is a reward value of $-1$ for encountering a cell with mud and a reward value of $-5$ for encountering a bomb. Reaching a goal or a bomb ends the agent's episode.

Each state $s$ is characterized by a feature mapping $\phi^{E_{\text{path}}}(s)$ which encodes the agent's location and direction, as well as the position of bombs, muds, and goals in the grid-world. In our environment, we consider grid-worlds of size $6 \times 6$, and each cell in the grid has a binary feature vector of length 7 as shown in Table 1. The first 4 features are a one-hot encoding representation of the agent's location and direction in the grid-world. The last 3 binary features represent the presence or absence of either a mud, bomb, or goal respectively at a cell. Consequently, the feature mapping $\phi^{E_{\text{path}}}(s)$ is of dimension $6 \times 6 \times 7$.

**TSP environment.** A task in the TSP environment is represented by a grid-world containing the agent and goal cells. Each possible configuration of a grid-world is associated with a state in the TSP environment MDP, $\mathcal{M}_{\text{tour}}$, similar to $\mathcal{M}_{\text{path}}$. The agent's action space $\mathcal{A} = \{\texttt{move}, \texttt{left}, \texttt{right}\}$ is defined the same as for $\mathcal{M}_{\text{path}}$. In this environment, the reward function $R^{E_{\text{tour}}}$ has a $+10$ reward value for completion of a successful tour, i.e., arriving back at the initial location after having visited all the goals in the grid-world. Similar to the shortest path environment, there is a reward value of $-1$ for each action performed by the agent. The agent's episode ends on the completion of a successful tour or after a certain time horizon.

Each state $s$ in the TSP environment is characterized by a feature mapping $\phi^{E_{\text{tour}}}(s)$ similar to the shortest path environment. We again consider grid-worlds of size $6 \times 6$, and each cell in the grid has a binary feature vector of length 6 as shown in Table 2. The first 4 features are a one-hot encoding representation of the agent's location and direction in the grid-world. The next feature captures the starting cell of the agent, which signals the final point of the tour. The last binary feature represents the presence or absence of a goal at a given cell. Hence, the feature mapping $\phi^{E_{\text{tour}}}(s)$ is of dimension $6 \times 6 \times 6$.

### D.2 Dataset Generation

Here, we outline the dataset generation process. We create separate training, validation, and test sets for both of our navigation environments. Further, optimal paths were computed for all the tasks in the training set for both environments. These are provided as demonstrations to the learner during the training phase. In the case of multiple optimal paths for a task, each optimal path was included as a unique demonstration.

| Agent facing North |
| --- |
| Agent facing South |
| Agent facing West |
| Agent facing East |
| Mud |
| Bomb |
| Goal |

Table 1: Shortest path task features

| Agent facing North |
| --- |
| Agent facing South |
| Agent facing West |
| Agent facing East |
| Start |
| Goal |

Table 2: TSP task features

---

**Algorithm 2** Scheduling Mechanism

---

1: **Initialization:** parameters a, b and total training epochs $N$.
2: **for** Epoch $e = 1, 2, \ldots, N$ **do**
3:     Curriculum strategy computes a preference over all demonstrations $\Xi$.
4:     Scheduling size is computed as $X = \begin{cases} b|\Xi| + \frac{e}{aN}(1-b)|\Xi| & \text{if } e < aN \\ |\Xi| & \text{otherwise} \end{cases}$
5:     The $X$ most preferred demonstrations are provided to the learner in random batches.

---

**Shortest path environment.** For the shortest path navigation tasks, we sample grid-worlds containing several muds and bombs, both in the range $\{0, \ldots, 12\}$. The agent's initial position and location of goals, muds, and bombs are all sampled at random without overlap. The training, validation, and test sets contain 100, 10, 30 grid-worlds respectively for each combination of muds and bombs, leading to datasets of sizes 16900, 1690, and 5070 respectively. Additionally, each dataset contains an equal percentage of grid-worlds with a single goal cell and with two goal cells.

**TSP environment.** For the TSP navigation tasks, we sample grid-worlds containing goal cells in the range $\{2, \ldots, 4\}$. The agent's initial position and location of goals are sampled at random without overlap. The training, validation, and test sets contain 2000, 100, 500 grid-worlds respectively for each unique number of goal cells in a task, leading to datasets of size 6000, 1500, and 300 respectively.

### D.3 Teacher's Difficulty Score

As explained in Sections 3 and 6, for the learner-centric setting we define the teacher's difficulty $\Psi^E(\xi)$ using a task-specific difficulty score.

**Shortest path environment.** For the shortest path tasks we define the following difficulty score:

$$\Psi^E(\xi) = \frac{\#goals \times \#optimal\_paths}{optimal\_reward}. \tag{7}$$

Intuitively, the difficulty score in Eq. (7) is proportional to the difficulty of a task as the greater the number of goals present and optimal paths, the more challenging the task is for the learner. Additionally, a higher optimal reward implies a shorter path to a goal that is less challenging for the learner.

**TSP environment.** For the TSP tasks we define the teacher's difficulty score as:

$$\Psi^E(\xi) = \frac{\#goals}{optimal\_reward - greedy\_gap}, \tag{8}$$

where *greedy gap* is defined as the difference in reward between the optimal tour and the greedy tour for the given task. In the greedy tour, the agent repeatedly navigates to the closest goal which has not been visited yet. Once all goals have been visited, the agent returns to its initial location. The greedy tour is not necessarily the optimal tour for a task.

Following a similar intuition as before, we see that the difficulty score of Eq. (8) is proportional to the difficulty of a task. The greater the number of goals and the lower the optimal reward, the greater the difficulty of the task for the learner. Further, tasks with a larger $greedy\_gap$ are more complex for the learner.

In both Eqs. (7) and (8) the denominator for the training tasks are linearly transformed to make all values $\geq 1$.

### D.4 Scheduling Mechanism

As commonly done in prior work [32, 69] when training neural networks using curriculum learning, we incorporate randomization in the training process for our CUR algorithm and its variants using a scheduling mechanism. Demonstrations of higher preference are prioritized at the beginning of

|  | **Input feature mapping** $6 \times 6 \times d$ |
|---|---|
| Convolution | Conv2D, kernel size = 3, padding = 1, $d \rightarrow 32$ |
|  | ReLU |
| Residual Block 1 | Conv2D, kernel size = 3, padding 1, $32 \rightarrow 32$ |
|  | ReLU |
|  | Conv2D, kernel size = 3, padding 1, $32 \rightarrow 32$ |
|  | ReLU |
|  | Conv2D, kernel size = 3, padding 1, $32 \rightarrow 32$ |
|  | ReLU |
| Residual Block 2 | Conv2D, kernel size = 3, padding 1, $32 \rightarrow 32$ |
|  | ReLU |
|  | Conv2D, kernel size = 3, padding 1, $32 \rightarrow 32$ |
|  | ReLU |
|  | Conv2D, kernel size = 3, padding 1, $32 \rightarrow 32$ |
|  | ReLU |
| Fully Connected | Linear, $6 \times 6 \times 32 \rightarrow 512$ |
| Fully Connected | Linear, $512 \rightarrow 256$ |
| Fully Connected | Linear, $256 \rightarrow 3$ |

Table 3: Network architecture

training, while during later stages, all demonstrations are provided with uniform probability to the learner.

In our experiments, we use a linear scheduling mechanism [69], where the first training epoch includes a fraction $b$ of the total demonstrations. The number of demonstrations included grows linearly every subsequent epoch such that by the time a fraction $a$ of the total epochs are completed, all the demonstrations are included. Algorithm 2 details the scheduling mechanism. The demonstrations in an epoch are provided to the learner in randomly ordered batches. In our experiments we set $a = 0.8$ and $b = 0.5$.

### D.5   Learner model

**Training hyperparameters.**   For both navigation environments, the learners were trained for 40 epochs with an initial learning rate of 0.01 and a batch size of 32 demonstrations. The learning rate was decayed by a factor of 0.5 after every 500 batches of demonstrations. The learning rate decay rule ensures the learning rate is consistent across the different curriculum algorithms since CUR and its variants utilize a different number of training tasks in each epoch due to the scheduling mechanism. Our models were trained on Nvidia Tesla V100 GPUs.

**Network Architecture**   The learner's neural network takes as input the feature mapping $\phi^E(s)$ of a state $s$. The dimension of the feature mapping is given by $6 \times 6 \times d$, where $d = 7$ for states in the shortest path navigation environment and $d = 6$ for states in the TSP navigation environment. In turn, the learner's neural network outputs a vector of size 3, which provides a probability distribution over actions after the softmax function is applied. The architecture of the neural network is provided in Table 3.

### D.6   Curriculum Visualization

In addition to the results presented in Section 6, we visualize the curriculum generated by our CUR algorithm for the shortest path and TSP environments in Figs. 8 and 9 respectively. In Figs. 8 and 9, the y-axis represents different features of the tasks provided to the learner, normalized in the range $[0, 1]$ and calculated as a moving average over the previous 100 batches. The x-axis represents the number of demonstrations provided to the learner.

**Shortest path environment.**   Fig. 8 shows that at the beginning of training, the CUR algorithm picks tasks with a fewer goals and a higher number of muds/bombs. We hypothesize that this teaches the agent how to avoid muds and bombs while navigating to a goal. During later stages

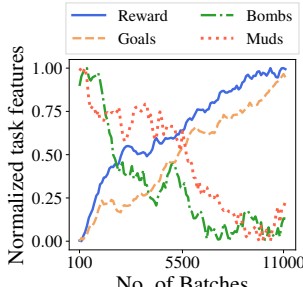

Figure 8: Shortest path environment curriculum visualization.

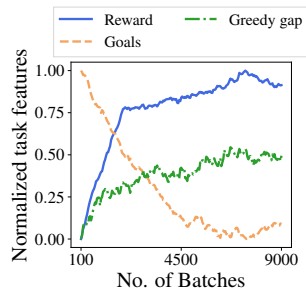

Figure 9: TSP environment curriculum visualization.

of training, CUR picks tasks with higher optimal rewards and a greater number of goals. Here we believe the agent is taught how to identify the path with maximum reward among all paths that lead to a goal. Essentially the learner is first taught the general navigation task followed by the most difficult concept of deciding the optimal path.

**TSP environment.** Fig. 9 illustrates that at the beginning of training CUR selects tasks with a greater number of goals, but with a low greedy gap. This would teach the learner the general navigation problem of visiting all goals. As training progresses, CUR picks tasks with a greater greedy gap. We hypothesize that these tasks teach the learner the most difficult concept of planning the optimal tour.