# OpenReview forum: "Curriculum Design for Teaching via Demonstrations: Theory and Applications"
_NeurIPS.cc/2021/Conference — NeurIPS 2021 Poster_

### Official Review · Reviewer_RaMd · 2021-07-14

**Rating:** 7
**Confidence:** 4

**Summary:**

This paper considers the problem of designing a curriculum of trajectories to teach an imitation learning agent. The authors propose to sequentially give trajectories that are both difficult (low probability) for the current learner and easy (high probability) for the demonstrator. Some theoretical justification is provided for why this approach works as well as some empirical results on simple domains. In general this paper seems novel, but there are many areas that deserve more clarification and it is unclear how far this idea can be taken given the strong assumptions.

**Limitations And Societal Impact:**

yes

**Main Review:**

Overall, I like this paper. The problem setting is interesting and the theoretical analysis is nice and some of the empirical results are compelling. However, there are three main weaknesses that I see that should be addressed:.

(1) The authors claim that batch curriculum approaches are not as good as interactive settings. However, the approach is not compared with batch curriculum approaches. The authors need to compare with a batch setting to justify why to select the trajectories sequentially. For example, the SCOT algorithm (citation [18]) uses a set cover approximation to design the entire curriculum offline. While this method was not originally designed for training BC agents, it can be used as a baseline in the MaxEnt IRL agent setting. It is unclear whether the proposed methods are that much better as they require access to the learner's policy at each iteration. It seems that an offline curriculum algorithm will be much more efficient as it only requires access to the MDP and the teacher's policy and does not require repeatedly probing the learner. However, I agree that there is a potential for the sequential approach to be better as it can adapt to the learning progress. This hypothesis needs to be rigorously tested by comparing the proposed approach with the prior work on batch teaching for LfD.

(2) Probing the learner's policy seems very difficult. The authors mention monte-carlo techniques but this would require repeatedly querying for actions at every state, if I understand correctly. Is there a possibility of devising a curriculum that doesn't require this many queries? See some ideas below. Also, it's unclear how much of an advantage it is to have access to the policy but not the internal parameters as in prior work [28]. What is the benefit of also having access to the internal parameters as done in prior work? I don't think this point needs to be solved, but more discussion of this limitation would strengthen the paper.

(3) Figure 5 shows that you don't need the hand-crafted difficulty score since you can just use the learner's difficulty. This seems to make the entire section 6 unnecessary. I'm not convinced that equation (4) is a good surrogate for teacher difficulty. In particular it seems that having more goals makes it easier not harder since there are more ways to solve the task near optimally.
Also the interpretation of the curriculum in Figure 6 seems iffy. The number of goals is included in the surrogate difficulty score f^E so it seems like that is the main reason later stages have more goals, not that this is actually a more difficult concept. I would recommend that the authors compare qualitatively with CUR-L to see if you get the same or different curricula if you don't use f^E.

I like the car driving example for visualizing and testing curriculum learning.

Clarity:
The authors often say that their method doesn't require access to the "learner's internal dynamics". This is incorrect and the wording should be changed. The internal dynamics are the way that the learner updates its model. This is assumed to be known. The unknown thing is the learner's parameters, not their internal dynamics. Just saying "internal parameters" would be much better and clearer.


Questions:
What happens with deterministic policies? How do you define/deal with 0/0 when selecting trajectories.

It would be good to relate this approach to prior work by Brown et al. on "Value Alignment Verification"? There they also seek an epsilon-approximate value difference. But look at a testing problem rather than a teaching problem. This enables them to verify the student policy is within epsilon of the teacher without requiring access to the entire policy of the learner as assumed in this work. Can something similar be done here?

Along similar lines, could ideas from Jacq et al. "Learning from a learner" allow curriculum design for a MaxEnt agent without needing to query for the entire policy?

Does the proposed approach work for continuous action spaces? What would be needed to extend to this setting?

Why have two separate theorems (1 and 3)? They say basically the same thing. If the proof techniques are different enough to justify different theorems it would be nice to give a proof sketch for how they are different. Or just merge them into one theorem.

Also, why is there no analogy to Thm 2 for BC? I think this is fine to leave for future work, but it would be beneifical to add a brief description of why this setting is so much more challenging than that for Thrm 2.

In figure 3b and d, why teach what is known initially later in training? Is this implicitly accounting and correcting for catastrophic forgetting?

The reward function still appears to be linear since there is an indicator for the conjunction, right?

The Ominicient teacher is not really ever explained in section 3.

How would this work for educational applications? There I'd assume that you know the teaching policy but can never really know the learner policy. This seems to make the proposed approach ill-suited since the experiments show that you can get around not knowing the teacher's policy, but need access to the full policy of the student.

---
Thank you for your extensive response. The authors have answered my questions and based on the response, I think the paper quality will improve.

---
Update: After a fruitful discussion with the other reviewers and the authors I am satisfied with the claims of novelty in the paper. I think these claims are nuanced and deserve better attention and I would like to see a more careful discussion of the differences between the submission and the blackbox approach proposed by Kamalaruban et al.. I think these points are easy to fix and will strengthen the paper. I would recommend that the paper be accepted.


**Time Spent Reviewing:**

6

---

> ### Author Response · Authors · 2021-08-09
> **Response to Reviewer RaMd**
>
> Thank you for carefully reviewing our paper! We greatly appreciate your feedback. Please see below our responses to your comments.
>
> -----
> **1. The authors need to compare with a batch setting**
> - Thank you for the feedback! As suggested, we have now done a few additional experiments with the SCOT algorithm as a baseline. We report on initial results in the tables below, where we compare CUR vs. SCOT. These initial findings suggest that the batch teaching algorithm is worse than our curriculum algorithm in terms of convergence. We will perform a more extensive study and report results in the revised paper.
> - We would also like to hear the reviewer’s thoughts on how we can better adapt the SCOT algorithm for our setting. For instance, there were different choices on how to implement the SCOT algorithm and we did try a few variants (e.g., when the batch selected by SCOT is finished, we continued with i.i.d. sampling). Another reason why the SCOT algorithm might be underperforming in these experiments is that the setting in Figure 3 corresponds to non-linear rewards (also, please see our response to comment 11 below).
> - The table below shows results in the context of Figure 3a where the MaxEnt-IRL learner’s initial knowledge is T0.
> | Teacher | t=0 | t=50 | t=100 | t=150 | t=200 |
> | ----- | --- | --- | --- | --- | --- |
> | CUR | -5.82 | 0.70 | 1.66 | 2.15 | 2.33 |
> | SCOT | -5.82 | -0.36 | 0.55 | 1.67 | 1.95 |
>
> - The table below shows results in the context of Figure 3c where the MaxEnt-IRL learner’s initial knowledge is T0-T3.
> | Teacher | t=0 | t=50 | t=100 | t=150 | t=200 |
> | ----- | --- | --- | --- | --- | --- |
> | CUR | -0.66 | 1.86 | 2.23 | 2.47 | 2.58 |
> | SCOT | -0.66 | 0.59 | 1.01 | 1.87 | 2.20 |
>
>
> **2.  Is there a possibility of devising a curriculum that doesn’t require this many queries?**
> - Based on the reviewer’s comment, we did an experiment to evaluate our curriculum algorithm when the learner’s policy is observed less frequently. In particular, we consider that the learner’s policy is observed only after every B time steps; for time steps in between, the curriculum algorithm will continue using the last observed policy. The CUR teacher in the paper corresponds to B=1, and we compare it with B=25 below.  The results show that CUR_B=25 performs quite well even though it gets to observe the learner’s policy very infrequently (i.e., a total of 9 times in this experiment).
> - The table below shows results in the context of Figure 3a.
> | Teacher | t=0 | t=50 | t=100 | t=150 | t=200 |
> | ----- | --- | --- | --- | --- | --- |
> | CUR | -5.82 | 0.70 | 1.66 | 2.15 | 2.33 |
> | CUR_B=25 | -5.82 | -0.01 | 1.37 | 2.05 | 2.20 |
>
> - The table below shows results in the context of Figure 3c.
> | Teacher | t=0 | t=50 | t=100 | t=150 | t=200 |
> | ----- | --- | --- | --- | --- | --- |
> | CUR | -0.66 | 1.86 | 2.23 | 2.47 | 2.58 |
> | CUR_B=25 | -0.66 | 1.83 | 2.23 | 2.43 | 2.53 |
>
> - In the above experiment, we assumed that the learner’s policy can be observed exactly. Another question is how to efficiently query the learner to approximate the learner’s policy. We discuss this point further in our response to comment 5.
>
> - Comparison with prior work [28]: The teaching setting in [28] makes use of the learner’s internal parameter and the dynamics of how the parameter gets updated for any given demonstration. This full knowledge about the learner makes the teaching setting in [28] more powerful in two ways: (a) a teacher has the ability to plan ahead and design more effective curriculum strategies; (b) a teacher has the ability to track the learner’s policy over time without repeated querying. On the other hand, our CUR teacher does not know the learner’s update rule. We further discuss this point in our response to comment 4.
>
> **3. Figure 5 shows that you don’t need the hand-crafted difficulty score**
> -  The reviewer’s assessment is correct. In the experiments of Section 6 and Appendix B.3, the CUR-L teacher performance is close to that of CUR, and the benefit of the hand-crafted difficulty score is not very significant. We also observed that the curricula generated by both teachers are quite similar, showcasing that the learner’s difficulty score plays a bigger role in these environments. However, as seen in the car driving setup of Section 5, the performance of CUR-L is worse compared to CUR. As future work, we plan to investigate these findings in more complex environments where the importance of the hand-crafted difficulty score f^E could be more significant. We believe the importance of Section 6 is in showcasing the effectiveness of our curriculum strategy in an MDP with a combinatorial state space.
>
> **4. Clarity: access to the learner’s internal dynamics**
> - We would like to clarify this issue in the context of two points: (a) learner’s update rule and (b) learner’s internal parameters.
> - Regarding the learner’s update rule: The work in [28] makes use of the learner’s update rule, i.e., how exactly the parameter gets updated for any given demonstration. In contrast, our CUR teacher does not know the learner’s update rule.
> - Regarding the learner’s internal parameters: Our CUR teacher only needs access to the learner’s current policy and not the exact internal parametric representation used by the learner. This allows our CUR teacher to be applicable for a broader class of learner models -- for a general learner model, one could infer the learner’s policy but might not be able to infer the internal representation.
>
> **5. Ideas from Brown et al. and Jacq et al.**
> - We thank the reviewer for the references. Below, we share some thoughts on how the techniques from these works could be utilized to alleviate the need to query for the “entire” policy.
> - The techniques from Brown et al. could be potentially useful in identifying parts of the state space where the learner’s policy deviates from the teacher. Then, the teacher can query the learner’s policy only in this region of high mismatch.
> - The techniques from Jacq et al. could be useful in two ways: (a) they have proposed an MLE method to approximate the learner’s policy from sampled demonstrations, and such a technique can be useful in our setting; (b) it would be interesting to investigate whether the teacher in our setting can be mapped to the “observer” in their setting, for instance, to track the learner’s policy over time without probing frequently.
>
> **6. What happens with deterministic policies?**
> - The proposed curriculum strategy can be applied to deterministic policies as well. Our CUR teacher only considers demonstrations that are realizable under the teacher’s policy. Hence, there is never a case of 0 appearing in the numerator. The case where the denominator is  (close to) 0 is less problematic as its corresponding trajectory would have a high rank within our curriculum strategy.
>
> **7. Does the proposed approach work for continuous action spaces?**
> - Following our reasoning above in comment 6, in principle, our curriculum strategy is applicable for continuous action spaces since we only consider demonstrations that are realizable under the teacher’s policy. However, in practice, our curriculum strategy has to be adapted in the following ways: (i) computationally efficient way of ranking over a large number of demonstrations, and (ii) sample efficient probing of the learner’s current policy.
>
> **8. Difference in Theorems 1 and 3**
> - Indeed, the proof techniques used for Theorems 1 and 3 are different. We will add a proof sketch noting these differences.
>
> **9. Add a brief description of why obtaining an analogy to Thm 2 for BC is much more challenging.**
> - Thanks for the suggestion! This challenge is also related to the distinction between the proof techniques used for Theorems 1 and 3. We will add a brief discussion on this in Section 4.2 of the revised paper.
>
> **10. In Figures 3b and d, why teach what is known initially later in training?**
> - We do not restrict our curriculum teacher to pick only the unknown tasks, and the tasks are picked based on the ranking strategy (Eq. 1). Indeed, we observe that the teacher is picking the initially mastered tasks later in training. As suggested, we also believe that it is implicitly identifying and correcting for catastrophic forgetting.
>
> **11. The reward function still appears to be linear since there is an indicator for the conjunction, right?**
> - We have a mistake in Figure 2, and the conjunction (HOV AND policy) feature is not part of the feature representation $\phi^E(s)$. More concretely, the $\phi^E(s)$ representation only contains the first eight individual features in the list. Therefore, the reward function is nonlinear w.r.t. the features. We will fix this issue in the revised paper.
>
> **12. The Omniscient teacher is not really ever explained in section 3.**
> - We will explain this in more detail in the revised paper. Please see our responses to comments 2 and 4 above.
>
> **13. Applying our approach to educational applications**
> - We share a few thoughts on how we could extend the proposed approach to educational applications. First, we will need to relax the assumption of knowing the learner’s exact policy and incorporate a sample efficient probing technique as part of the teaching algorithm. Here, the ideas discussed in our responses to comments 2 and 5 will be useful. Second, for more complex application domains with large state/action spaces, it might be ineffective to represent the learner’s “knowledge” through a fine-grained policy. In this context, it will be useful to develop probing techniques that enable us to directly infer the difficulty scores over tasks needed for our curriculum strategy.
>
> -----
>
> We hope that our responses can address your concerns and are helpful for improving your rating. If you have any other comments or feedback, please let us know! We will be happy to provide further responses. We are looking forward to hearing back from you! Thank you again for the review.

---

> > ### Comment · Reviewer_RaMd · 2021-08-17
> > **response**
> >
> > Thank you for your response.
> >
> > I think the extra experiments and clarifications above significantly strengthen the paper. Regarding the comparison to the batch setting, it seems that one possible way of comparing batch with sequential is to only allow the sequential method the same number of queries as the batch. Since the batch is designed to fully teach the desired reward function in one batch, it seems that the sequential method's performance should be calculated after being allowed |Batch| queries. However, I think the initial experiment proposed by the authors is also useful to have as a comparison.
> >
> > I am satisfied with the responses. I still think requiring the entire policy is a strong requirement, but I appreciate the novelty of the results and theory and I think the new experiment is promising that shows that the policy does not have to be sampled for each query. I am increasing my score for the paper.

---

> > > ### Author Response · Authors · 2021-08-22
> > > **Thank you for the feedback**
> > >
> > > We thank the reviewer for the valuable and constructive feedback. We will perform a more extensive study in comparing our algorithms to the batch setting. We will include additional results and discussions in the revised paper. Thank you again for the review!

---

### Official Review · Reviewer_3tg2 · 2021-07-16

**Rating:** 6
**Confidence:** 3

**Summary:**

The authors introduce a new method for imitation learning, and more specifically, for selecting an expert demonstration that is maximally useful for the learner.

**Limitations And Societal Impact:**

The authors did not properly explain the limitations of this work and neither the societal impact. Both were only mentioned off-hand in the conclusion.
Here are some suggestions:
- Instead of "our approach could be beneficial for education applications", you could write something more concrete like "We believe that this method can be employed by MOOC platforms to gauge which order of courses would be most beneficial for the students..."
- As for limitations, I would recommend to put the contribution of this work into perspective to other contemporary works and add a few sentences a la "we only experimented in a limited grid world setting, we only experimented with a low number of random seeds, we assume that we have unlimited access to a teacher,  etc"

**Main Review:**

**Originality:**

The method that is introduced here is not inherently bad but it only appears to work on small isolated grid world problems in which all states and actions can be enumerated. I believe that with automatic curriculum methods like DAGGER [1] and AGGREVATE [2] (which work in continuous state/action spaces), and astounding results such as "Learning to drive in a day" [3], the relevancy of this contribution is questionable at best. I guess the only way to convince me otherwise would be to include experiments in continuous state/action space but I don't think this can be part of the discussion period.

The authors also omitted a related literature section and instead list a few related works in the introduction, which does not help separate this work clearly from its predecessors.

I also feel like Behavior Cloning is treated as state of the art for imitation learning and from my best understanding that is the most naive possible approach. Some more contemporary baselines are DAC [4], TCN [5], and VIRL [6] that learn trajectory embeddings of the expert which could be used for comparison, rather than the product of single-state differences, which is one of the proposed metrics in this work.

**Quality:**

1. In the car environment, the learned curriculum is visualized (Fig.3 and 4, b & d) and there are artifacts that need to be explained: (a) why does the learner with initial knowledge of T0 go back to learning T0 after T=150, why does it ignore T1 instead? (b) what is the "time" axis? Is that supposed to be environment interactions? (c) Why does the agent with knowledge of T0-T3 gradually resume these tasks? (d) ...and why does it initially ignore T4 and T5? (e) How many seeds were used to train the agents in Fig. 3/4?
2. All that the experiments in the 6x6 grid world show is that the agent learns to maximize equation 4, i.e. create more goals and fewer bombs/mud and therefore create easier and easier environments, which is backed up by Figure 5c. What are these experiments supposed to show? The agent has not acquired any resilience to bombs and mud but rather found a way to remove them from the environment altogether.

**Clarity:**

The writing is mostly clear.
1. A dedicated "Related Literature" section would be good
2. line 80 $\Theta$ is not defined
3. line 82 should be an equation and not line-broken
4. Algorithm 1 can be cut because it's trivial or can be explained in 2 sentences.
5. The entirety of sections 4(.0) and 4.1 is pointless in my eyes because it assumes comparability of the agent parameters $\theta$ to the expert parameters $\theta^*$ and that is not a realistic or even plausible assumption, given that an expert can be of a completely different structure (e.g. a human demonstrating to a neural network)
6. Fig. 1 and 2 are missing an explanation of what "HOV" is. (I'm a native English speaker and I had to google that.)
7. Fig. 6 could use a legend of what the different symbols mean. It's clear from the text but Fig. 6a would be easier to parse with one.

**Significance:**

As outlined above, I do not believe this method is pushing the state-of-the-art.

**Nitpicks and Questions:**

1. The authors keep repeating the phrase "under minor technical conditions" 3 times in the paper and I have no idea what this is supposed to mean.

**References:**

- [1] Ross, Stéphane, Geoffrey Gordon, and Drew Bagnell. "A reduction of imitation learning and structured prediction to no-regret online learning." Proceedings of the fourteenth international conference on artificial intelligence and statistics. JMLR Workshop and Conference Proceedings, 2011.
- [2] Ross, Stephane, and J. Andrew Bagnell. "Reinforcement and imitation learning via interactive no-regret learning." arXiv preprint arXiv:1406.5979 (2014).
- [3] Kendall, Alex, et al. "Learning to drive in a day." 2019 International Conference on Robotics and Automation (ICRA). IEEE, 2019.
- [4] Kostrikov, Ilya, et al. "Discriminator-actor-critic: Addressing sample inefficiency and reward bias in adversarial imitation learning." arXiv preprint arXiv:1809.02925 (2018).
- [5] Sermanet, Pierre, et al. "Time-contrastive networks: Self-supervised learning from video." 2018 IEEE international conference on robotics and automation (ICRA). IEEE, 2018.
- [6] Berseth, Glen, and Christopher J. Pal. "Visual Imitation Learning with Recurrent Siamese Networks." arXiv preprint arXiv:1901.07186 (2019).

**Time Spent Reviewing:**

7

---

> ### Author Response · Authors · 2021-08-10
> **Response to Reviewer 3tg2**
>
> Thank you for your comments and suggestions. Please see below our responses to your comments. We have clarified several misunderstandings regarding the experimental evaluation and the significance of our theoretical contributions. We request the reviewer to consider our response as well as the feedback of other reviewers. We hope that our responses can address your concerns and are helpful for improving your rating.
>
> -----
> **1. “the method that is introduced here is not inherently bad but it only appears to work on small isolated grid world problems”**
> - We want to clarify the reviewer’s misunderstanding about the experimental setup and different environments considered in the paper. In particular, for the experiments in Section 6 and Appendix B.3, the agent is learning a multi-task neural policy for solving any navigation task provided as input -- crucially, the state space corresponds to all possible configurations of these navigation grids, and the state space size is combinatorial in terms of the number of objects. For instance, consider the environment in Section 6: here, the state space size in the MDP is over 10^10, corresponding to different ways of placing bombs/muds/goals. This state space is intractably large to enumerate.
>
> **2. “all that the experiments in the 6x6 grid world show is that the agent learns to maximize equation 4, i.e. create more goals and fewer bombs/mud and therefore create easier and easier environments, which is backed up by Figure 5c”**
> - We believe there is a serious misunderstanding regarding our experimental evaluation. We consider different curriculum strategies for training an agent, and these strategies are evaluated based on the performance of the agent’s policy on a fixed set of test tasks. Crucially, the same test set is used for comparing different curriculum strategies.
> - More concretely, as stated in line 318, Figure 5b shows the performance of the agent’s trained policies in terms of expected reward on the test set (y-axis) across the training phase (x-axis). Furthermore, as stated in lines 321-324, Figures 5c and 6 visualize the curriculum in terms of tasks selected during training (however, the performance is measured on the test set as mentioned above).
>
> **3. “the entirety of sections 4(.0) and 4.1 is pointless in my eyes”**
> - We respectfully disagree with the reviewer. The curriculum learning paradigm has been very popular for applications of supervised learning and sequential decision-making; however, there is a lack of theoretical tools for analysing and developing principled curriculum methods. In the past few years, there have been several exciting theoretical results on curriculum design in the context of supervised learning -- for instance, [Weinshall et al. 2018] and [Weinshall and Amir 2018] introduced task-difficulty scores to develop a theory of curriculum learning for linear regression and binary classification settings. There has also been a lot of interest in developing a theory of curriculum design in sequential decision-making settings -- for instance, see [Brown and Niekum 2019] and [Kamalaruban et al. 2019]. Our work builds on the theoretical idea of using “the differential effect of difficulty scores” from [Weinshall et al. 2018], and introduces notions of task-difficulty scores that are useful in designing curriculum for imitation learning agents (see Theorems 1 and 3).
> - As theoretical contributions, our work improves upon the existing theory on curriculum design for imitation learning from [Brown and Niekum 2019] and [Kamalaruban et al. 2019] in the following ways. In contrast to [Brown and Niekum 2019], we develop an interactive teaching algorithm that is able to utilize the learner’s current policy to design a curriculum adapted to the learner’s progress. In contrast to [Kamalaruban et al. 2019], our teaching algorithm does not require knowledge of the learner’s update rule. Furthermore, we provide analysis for two popular imitation learner models: the CrossEnt-BC learner model and the MaxEnt-IRL learner model (the results in [Kamalaruban et al. 2019] didn’t consider CrossEnt-BC). Another significant contribution of our work is that we are able to prove the convergence of the MaxEnt-IRL learner when trained using our curriculum strategy (see Theorem 2).
>
> - Now, coming back to the reviewer’s specific points: (1) the above-mentioned state-of-the-art works, [Weinshall et al. 2018; Weinshall and Amir 2018; Brown and Niekum 2019; Kamalaruban et al. 2019], also incorporate both the teacher’s and the learner’s parameters to develop a theory for curriculum learning; and (2) our curriculum strategy, Eq. 1, which we developed using insights from our theoretical analysis in Section 4, can indeed be applied in a setting where the teacher and the learner have a different representation structure.
>
> [Weinshall et al. 2018] Weinshall et al. “Curriculum learning by transfer learning: Theory and experiments with deep networks.” ICML 2018.
>
> [Weinshall and Amir 2018] Weinshall and Amir. “Theory of curriculum learning, with convex loss functions.” arXiv 2018.
>
> [Brown and Niekum 2019] Brown and Niekum. “Machine teaching for inverse reinforcement learning: Algorithms and applications.” AAAI 2019.
>
> [Kamalaruban et al. 2019] Kamalaruban et al. “Interactive teaching algorithms for inverse reinforcement learning.” IJCAI 2019.
>
> **4. The relevancy of our contributions in the context of suggested references and other imitation learning methods**
> - We thank the reviewer for the references provided, and we will expand on related works in the revised paper to make the relevancy of our contributions more clear. We believe the contributions of our work are complementary to the works suggested by the reviewer. As we discussed in comment 3 above, one of our main contributions is developing a theoretical understanding of curriculum design for imitation learning agents. While we focus on specific learner models for developing the theory, the insights from our work could be potentially useful in analysing and developing curriculum strategies for more advanced learner models in future works.
>
> **5. “The authors also omitted a related literature section and instead list a few related works in the introduction”**
> - Thank you for the suggestion. We will incorporate a dedicated “Related Literature” section in the revised paper.
>
> **6. “In the car environment, the learned curriculum is visualized (Fig.3 and 4, b & d) and there are artifacts that need to be explained”**
> - We will provide more explanations about the figures in the revised paper. We note that our curriculum teacher picks tasks based on the ranking strategy and is not restricted to pick any specific tasks (e.g., unknown tasks only). Indeed, we observe that the teacher is picking the initially mastered tasks later in training. Mathematically, this phenomenon could occur due to the following reasons: (i) specific trajectories could have a higher preference even for already mastered tasks (as per Eq. 1), and (ii) specific tasks could bring more incremental benefit in terms of the agent’s performance (as per Eq. 3).
> - In particular, we will expand on the following answers: (a) at a later stage of training (t=150 onwards), the learner has nearly mastered all tasks, and the CUR teacher’s ranking strategy could pick previously mastered tasks (because of ranking preference over specific trajectories or higher learning benefit); (b) the time axis represents time steps t as defined in Algorithm 1; (c) when not observing certain tasks for a while, the agent’s performance on these tasks could degrade which is identified and corrected for by the curriculum; (d) the skills required for T4 and T5 are present in other tasks, e.g., avoiding grass in T6, so it could be more effective to demonstrate other tasks which train more skills; (e) as mentioned in line 261, 10 random seeds were used to train the agents for the reward convergence plots in Figures 3 and 4.
>
> **7. Comments on clarity and suggestions for improving the writing**
> - Thanks for the comments and suggestions. We will incorporate the following into the revised paper: (a) add a dedicated related work section, (b) add the suggested corrections in lines 80 and 82, (c) clarify the term “HOV”, (d) add a legend in Figure 6, and (e) add technical conditions on eta in the main text.
>
> **8. “The authors did not properly explain the limitations of this work and neither the societal impact. Both were only mentioned off-hand in the conclusion.”**
> - Thank you for your suggestions on the writeup for limitations and societal impact. We will update the writing in the revised paper.
>
> -----
> We hope that our responses can help address your concerns. We have also provided more experimental details, empirical results, and analysis in the appendix of the supplementary material. If you have any other comments or feedback, please let us know! We will be happy to provide further responses. We are looking forward to hearing back from you. Thank you again for the review.

---

> > ### Comment · Reviewer_3tg2 · 2021-08-16
> > **Response**
> >
> > **Re 1, “the method that is introduced here is not inherently bad but it only appears to work on small isolated grid world problems”**
> >
> > The fact that there can be different items in a grid cell, doesn't make the environment any less gridworld.
> > But maybe I'm missing some points here. Can you list out the assumptions and constraints on the environment, please? I.e. Does this work in a non-discrete environment, does this work with stochastic state transitions, etc.? And in addition to that, how often does the expert get sampled in the CUR algorithm in any given episode?
> >
> > **Re 2. “all that the experiments in the 6x6 grid world show is that the agent learns to maximize equation 4, i.e. create more goals and fewer bombs/mud and therefore create easier and easier environments, which is backed up by Figure 5c”**
> >
> > In Fig. 5c, why don't the values start all at 0 or all at 0.25? What bias is baked into the environment?
> >
> > Also, yes, I understand that Fig. 5b is reporting performance on the test set, not training set. But is equation 4 not effectively being minimized by the agent? Is that not what Fig. 5c expresses? The agent samples more positive reward signals and fewer negative reward signals from the teacher/environment, correct?
> >
> > **Re 3. “the entirety of sections 4(.0) and 4.1 is pointless in my eyes”**
> >
> > Can you show any papers that were accepted at NeurIPS 2020 or 2019 that have a similar profile/positioning/contribution?
> >
> > But also thank you, this section should be included in the paper. I'll have a look at some of these works.
> >
> > ---
> >
> > Thanks for the other comments and clarifications. I'm already learning towards increasing my score by 1-2 points and I'll consider another increase depending on the answers to my questions above.

---

> > > ### Author Response · Authors · 2021-08-22
> > > **Response to Reviewer 3tg2 (Part 2)**
> > >
> > > Thank you for the follow-up comments. Please see below our responses.  We hope that our responses can address your remaining concerns and are helpful for improving your rating. If you have any other comments or feedback, please let us know! We will be happy to provide further responses. Thank you again for the review.
> > >
> > > -----
> > >
> > > **9. The fact that there can be different items in a grid cell, doesn't make the environment any less gridworld ... Can you list out the assumptions and constraints on the environment, please? ... how often does the expert get sampled in the CUR algorithm in any given episode?**
> > >
> > > - **How often does the expert get sampled in the CUR algorithm?:** First, we would like to clarify how different curriculum algorithms are evaluated in Section 6. We begin by creating a fixed pool of grid-world tasks and split them into training and test sets. Each task corresponds to a grid-world with the location of goals, muds, bombs, and an initial location of the agent; for each of these tasks, we obtain an expert trajectory corresponding to an optimal navigation path in that grid-world. The details about dataset generation are in Appendix B.2 lines 517-523 --  in our experiments, we had $16900$ and $5070$ number of tasks in the training and test sets respectively. For the evaluation, all the curriculum algorithms are given the following: (i) the training set (i.e., $16900$ grid-world tasks along with expert trajectories) and (ii) a training budget, i.e., the number of episodes in the training process where each episode corresponds to one such task. The goal is to learn a multi-task neural policy that can generalize to new unseen tasks -- the performance is measured w.r.t. the tasks in the test set. These algorithms only differ in their curriculum (i.e., how they prioritize and rank these tasks for training). Fig. 5b shows the performance of different curriculum algorithms and can be interpreted as follows: (i) a point on the x-axis (Time t) corresponds to the number of training episodes where each episode is a task from the training set, (ii) a point on the y-axis (Expected Reward) is the performance of the learnt neural policy on tasks in the test set, and (iii) four different curriculums are evaluated (CUR, CUR-L, CUR-T, AGN). In Section 6, the expert trajectories are used only in generating the dataset, and, afterwards, the curriculum algorithms do not have access to the expert. For instance, the CUR algorithm in Fig. 5b picks tasks using Eq. 1 as follows: (a) the numerator $\Psi^L_t$ is based on the learner’s policy that is being learnt, and (b) the denominator $\Psi^E$ is based on $f^E$ as there is no access to the expert.
> > >
> > > - **Shortest path and shortest tour environments (Section 6 and Appendix B.3 respectively):** Building on the above discussion, we would like to highlight that the experiments in Section 6 and Appendix B.3 are for a multi-task learning setting -- the agent is learning a neural policy that can take any task as input (a grid-world configuration) and can compute a navigation path for the task. This is in contrast to an isolated grid-world (i.e., a single task) that can typically be solved using a tabular state space representation. In our experiments, the state space size in the MDP is intractably large for tabular representation and requires a neural representation. More concretely, we consider a convolutional neural network which takes as input a $6 \times 6 \times 7$ tensor, where each cell in the grid-world has a $7$ bit binary feature vector, and the network outputs a probability distribution over the actions. These environments represent a discrete combinatorial problem in a multi-task learning setting.
> > >
> > > - **The assumptions and constraints on the environment:** When generating the datasets for experiments in Section 6, we considered a few minimal constraints as follows: (a) a goal must be reachable from the agent’s initial location, and (b) the agent cannot be initially located on a goal. In the results reported in Fig. 5, the transitions are deterministic; in addition, we also evaluated these algorithms on a stochastic variant of the environment where each action is randomly flipped with another action with a probability p. That is, the results in Fig. 5 are for p=0; for non-zero values of p=0.05 and p=0.10, the results in terms of performance gap across algorithms are quantitatively very similar. In Section 5, the results are based on a stochastic environment which was taken from [Kamalaruban et al. 2019] to compare with their algorithm. In future work, it would be interesting to apply these algorithms in environments with continuous state/action spaces -- please see our response 7 to Reviewer RaMd where we have discussed a few points on the applicability of our approach to this setting.
> > >
> > > **10. In Fig. 5c, why don't the values start all at 0 or all at 0.25? What bias is baked into the environment?... But Is equation 4 not effectively being minimized by the agent? Is that not what Fig. 5c expresses?**
> > >
> > > - **Different curriculum algorithms in Section 6:** First, we would like to clarify how different curriculum algorithms in Section 6 work. More concretely, we have the following: (i) the CUR algorithm picks tasks from the training set using Eq. 1 where the numerator is $\Psi^L_t$ and the denominator is set to $f^E$ (i.e., $\frac{\Psi^L_t}{f^E})$; (ii) the CUR-L algorithm picks tasks from the training set using Eq. 1 where the numerator is $\Psi^L_t$ and the denominator is set to 1 (i.e., $\frac{\Psi^L_t}{1})$; (iii) the CUR-T algorithm picks tasks from the training set using Eq. 1 where the numerator is set to 1 and the denominator is set to $f^E$ (i.e., $\frac{1}{f^E})$; (iv) the AGN algorithm picks tasks i.i.d. Importantly, all these algorithms pick tasks from a fixed training set, as we mentioned in our response 9 above.
> > >
> > > - **Is equation 4 not effectively being minimized by the agent? Is that not what Fig. 5c expresses?:** As discussed above, CUR-T minimizes Eq. 4 and ignores the learner’s difficulty score $\Psi^L_t$; CUR also incorporates the learner’s difficulty score $\Psi^L_t$. As the convergence plot in Fig. 5b shows, CUR and CUR-L outperform CUR-T, thereby signifying the importance of the learner’s difficulty score.  The curriculum of CUR in Fig. 5c is in fact dominated by the learner’s difficulty score; also, note that the curriculum of CUR-T (that solely optimizes Eq. 4) would be expected to have “rewards” decrease over time. We will add additional curriculum plots in the revised paper to further highlight this point.
> > >
> > > - **In Fig. 5c, why don't the values start all at 0 or all at 0.25?** Fig. 5c visualizes the task features as picked by the CUR algorithm at a given time $t$. The graph plots the normalized task features, such as reward, goals, etc., during the training process. Fig. 5c curriculum visualization is analogous to the curriculums shown in Figs. 3b, 3d, 4b, 4d of Section 5. The initial values simply correspond to the tasks picked from the training set at time $t=1$ by the curriculum algorithm. There is no specific bias baked into the environment, and different curriculum algorithms have different curriculum curves along with different initial values. We will further clarify this point in the revised paper.
> > >
> > > **11. Can you show any papers that were accepted at NeurIPS 2020 or 2019 that have a similar profile/positioning/contribution?**
> > >
> > > - Our work is broadly related to the topics of curriculum learning and machine teaching.  Below, we list a few recent papers on these topics from NeurIPS and other AI/ML venues. From this list, the most closely related papers to our work are [Brown and Niekum 2019], [Kamalaruban et al. 2019], and [Weinshall et al. 2018] -- we have provided a more detailed comparison w.r.t. these papers in our responses.
> > >
> > > [Zhou et al. 2020] Zhou et al. “Curriculum learning by dynamic instance hardness.” NeurIPS 2020.
> > >
> > > [Sinha et al. 2020] Sinha et al. “Curriculum by smoothing.” NeurIPS 2020.
> > >
> > > [Nabli and Carvalho 2020] Nabli and Carvalho. “Curriculum learning for multilevel budgeted combinatorial problems.” NeurIPS 2020.
> > >
> > > [Feng et al. 2020] Feng et al. “A novel automated curriculum strategy to solve hard sokoban planning instances.” NeurIPS 2020.
> > >
> > > [Dennis et al. 2020] Dennis et al. “Emergent complexity and zero-shot transfer via unsupervised environment design.” NeurIPS 2020.
> > >
> > > [Peltola et al. 2019] Peltola et al. “Machine teaching of active sequential learners.” NeurIPS 2019.
> > >
> > > [Tschiatschek et al. 2019] Tschiatschek et al. “Learner-aware teaching: Inverse reinforcement learning with preferences and constraints.” NeurIPS 2019.
> > >
> > > [Haug et al. 2018] Haug et al. “Teaching inverse reinforcement learners via features and demonstrations.” NeurIPS 2018.
> > >
> > > [Zhou et al. 2021] Zhou et al. “Curriculum learning by optimizing learning dynamics.” AISTATS 2021.
> > >
> > > [Wu et al. 2021] Wu et al. “When do curricula work?” ICLR 2021.
> > >
> > > [Rakhsha et al. 2020] Rakhsha et al. “Policy teaching via environment poisoning: Training-time adversarial attacks against reinforcement learning.” ICML 2020.
> > >
> > > [Brown and Niekum 2019] Brown and Niekum. “Machine teaching for inverse reinforcement learning: Algorithms and applications.” AAAI 2019.
> > >
> > > [Kamalaruban et al. 2019] Kamalaruban et al. “Interactive teaching algorithms for inverse reinforcement learning.” IJCAI 2019.
> > >
> > > [Weinshall et al. 2018] Weinshall et al. “Curriculum learning by transfer learning: Theory and experiments with deep networks.” ICML 2018.
> > >
> > > [Zhou and Bilmes 2018] Zhou and Bilmes. “Minimax curriculum learning: Machine teaching with desirable difficulties and scheduled diversity.” ICLR 2018.
> > >
> > > -----
> > >
> > > We hope that our responses can help address your concerns. If you have any other comments or feedback, please let us know! We will be happy to provide further responses. Thank you again for the review.

---

> > > > ### Comment · Reviewer_3tg2 · 2021-08-30
> > > > **Raising score 2-> 6**
> > > >
> > > > Thanks for the detailed response. In response, I have raised my score.

---

> > > > ### Author Response · Authors · 2021-08-31
> > > > **Thank you for the feedback (Reviewer 3tg2)**
> > > >
> > > > We thank the reviewer for their feedback. We are glad that our responses were helpful in improving your rating. Thank you again for the review.

---

> ### Author Response · Authors · 2021-08-26
> **Re: Response to Reviewer 3tg2**
>
> We thank the reviewer again for their comments. We sincerely hope that our detailed responses have addressed your concerns and are helpful for improving your rating. If you have any other comments or feedback, please let us know! We will be happy to provide further responses. Thank you again for the review.

---

### Official Review · Reviewer_Ecwk · 2021-07-17

**Rating:** 7
**Confidence:** 2

**Summary:**

This paper considers the problem of providing experts demonstrations
to a learner, modeled as a Maximum Entropy IRL or a Cross Entropy
Behavioral Cloning algorithm, in order for them to achieve a target
performance on some task. The key contribution of the paper is to
provide a demonstration ranking strategy which can be greedily
utilized to define a curriculum. This strategy resulting in
convergence for both types of learners. Importantly, and in contrast
to prior work, this approach can be done in a manner that does not
explicitly see the learner's internal dynamics.  Moreover, in the case
of the maximum entropy IRL (with deterministic dynamics), the prove
that the convergence is logarithmic. Finally, they show the efficacy
of this approach in various domains.



**Limitations And Societal Impact:**

This approach extends the theoretical understanding of teaching to existing algorithms and potentially enable new forms of human robot interaction. None of these are particularly concerning and I would argue this work is a net positive impact.

Regarding the limitations, the primary limitation (acknowledged in the conclusion) is the sharpness of the convergence bounds for cross entropy and extending to more general dynamics. That said, I feel that it is perfectly reasonable to leave these as future work.

**Main Review:**

The considered problem of efficiently teaching common learning from
demonstration algorithms is certainty interesting and has a number
of theoretical and practical applications. For example, there is a
large body of literature of treating humans as maxEnt IRL learners
and optimizing the informativeness of demonstrations, particularly
in the context of human robot collaboration.

The key contribution of this paper, the ranking function, is fairly
simple to compute both directly given the learner's internal state
as well as empirically using monte carlo methods. As such, this
work opens up several possible applications compared to prior work.

An interesting feature, although whether it is a positive or negative
feature is another question, is that the convergence guarantees are
agnostic to the particular MDP being considered. One would naturally
expect that some MDPs are "easier" than others, but this is not
reflected in the bounds. An MDP sensitive analysis while more
difficult, may give a much sharper characterization of the learning
curves observed in practice.

My primary concern with the paper was the simple setting of
deterministic dynamics for maxEnt considered (although weakened in the
experiments). While the convergence guarantees are fairly strong, its
unclear to me if this should be expected to hold in more general
dynamics or is a special feature of the deterministic setting.

That said, the empirical evaluation seems to suggest that the
curriculum really does make a difference compared to an agnostic
teacher which is un-aware of the trace level difficultly.

## Minor points
Unless I'm misunderstanding, t, seems to be used in multiple ways,
both as the demonstration index and the time within an episode.

Should the footnote on the bottom of page 3 point to section 4?

The technical details of eta would really be nice in the main draft.

**Time Spent Reviewing:**

4

---

> ### Author Response · Authors · 2021-08-09
> **Response to Reviewer Ecwk**
>
> Thank you for carefully reviewing our paper! We greatly appreciate your feedback. Please see below our responses to your comments. We hope that our responses can address your concerns and are helpful for improving your rating or confidence score. If you have any other comments or feedback, please let us know! We will be happy to provide further responses. We are looking forward to hearing back from you!
>
> -----
> **1. Convergence guarantees are agnostic to the particular MDP being considered**
> - Thanks for the question! In the revised version of the paper, we will expand the presentation of Theorem 2 and provide more insights of the terms that appear in the $O(\cdot)$ notation. However, we agree with the reviewer that our analysis is not designed to specifically capture the MDP properties and our focus was primarily on the rate of convergence. It is definitely a very interesting direction for future work to develop instance-specific convergence guarantees that can account for the MDP properties. We will add this discussion in the revised version.
>
> -----
> **2. While the convergence guarantees are fairly strong, its unclear to me if this should be expected to hold in more general dynamics or is a special feature of the deterministic setting.**
> - It is indeed an important research question to extend the convergence guarantees for the general dynamics. Our initial efforts to tackle this question suggest that it is non-trivial to extend the analysis from deterministic dynamics to general dynamics. In particular, for the general dynamics, one needs to consider the MCE-IRL learner model instead of the MaxEnt-IRL learner model and account for the stochasticity of sampled trajectories for any initial starting state. We will add a discussion around these challenges in the revised version of the paper.
> - Nevertheless, we believe that our theoretical analysis for the simplified settings and experimental evaluation for the general settings provide an important step towards developing principled curriculum design approaches for imitation learning agents.
>
> -----
> **3. Minor points**
> - Thank you for the suggestions. We will add the technical details of eta in the main draft. Further, we will expand the discussion in Sections 4.1 and 4.2 to provide more insights about the theorems.
> - We will fix other minor points in the revised version of the paper.
>
> -----
> We have also provided more experimental details, empirical results, and analysis in the appendix of the supplementary material. If you have any other comments or feedback, please let us know! We will be happy to provide further responses. Thank you again for the review!

---

### Official Review · Reviewer_kSL2 · 2021-07-19

**Rating:** 5
**Confidence:** 4

**Summary:**

The paper investigates approaches to curriculum design for teaching via demonstrations, unifying two models in the literature. Some theoretical results are proven and experiments show the effectiveness of the approach.

**Ethical Concerns:**

None.

**Limitations And Societal Impact:**

No limitations were discussed. Societal impact was not discussed.

**Main Review:**

The paper presents a unified approach to curriculum design for MaxEnt-IRL and CrossEnt behavioral cloning. The primary contributions appear to be a teaching approach that does not require direct access to the learner's internal parameters, theoretical results regarding convergence rate, and modest sized empirical demonstrations of effectiveness.

Positive aspects:
- The problem of curriculum design is interesting and important, and compelling theoretical results are relatively limited at this time.
- The proposed approach based on difficulty scores is interesting.
- The empirical results are nice, if a bit modest in scope.

Weaknesses:
- The review of literature was somewhat limited. In particular, there are several papers by Mark Ho and Michael Littman and colleagues that seem relevant.
- The motivation for difficulty scores was weak. In particular, it seems connected to the broader class of approaches that trade off effort of the teacher and effect for the learner that appear in probabilistic formulations.
- The paper is rather niche. The theoretical results appear to be limited to the two approaches considered, and not general to curriculum learning. The empirical demonstrations are on small scale gridworld style problems. The exposition does not flow particularly well in that there are terms used without introduction, motivations for choices early in the paper are explained later, etc.
- Although the paper is billed as being about curriculum learning theory and applications, it is not general enough to live up to these promises.

Detailed comments:
- "much less work is done from the teacher’s point of view to reduce the number of demonstrations required to achieve the learning objective" There is a literature on Teaching dimension that is relevant. Also see Optimal Cooperative Inference.
- There is relevant work by Mark Ho, Michael Littman and others that does not appear to be reviewed.
- "We define difficulty scores for any demonstration" Please define terms such as "difficulty scores" when they are introduced.
- "We then study the differential effect of the difficulty scores" What does this mean?
- "We adapt our curriculum strategy to utilize domain knowledge, in the form of task-specific difficulty scores, when the teacher’s optimal policy is unknown." What does this mean? What constitutes domain knowledge? What are difficulty scores? Teacher's policy unknown to whom?
- ", the teacher could approximately infer the policy πLt by probing the learner and using Monte Carlo methods" Probing the learner?
- "Leads to CUR teaching algorithm" How does it lead to it?
- Self-curriculum is not defined.
- I had trouble understanding the feasible set introduced on lines 142-143. The difficulty values are the same or different from difficulty scores? What type are they?
- "In our experiments we relax the simplification" What does this mean?
- Rather than using the phrase "differential effect" it would be clearer to just say what the effect is. Differential effect is not that helpful for the reader to whom the details have not been explained.
- " The result is analogous to that obtained for linear regression" Analogous how?
- I do not support the choice to put the technical details on nu_t in the appendix. One could at least give the reader a flavor.
- "Theorem 1 motivates our curriculum strategy in Eq." How? Also, the ordering of the exposition is not ideal, given that Eq 1 appears much earlier.
- In theorem 3, what does it mean for the convergence rate to be "roughly" something?
- The use of a scheduling mechanism is fine in principle. But it is important to assess all algorithms on the same footing, so we can differentiate between what is due to the algorithm and what is due to the training approach. It is not ideal to put details relevant to understanding performance in the appendix.
- " This clearly shows the effectiveness of our proposed ranking strategy that combines the two difficulty scores" This does not seem quite accurate. The result shows that both difficulty scores matter. I do not believe we can tell what it says about the particular manner of combining them.
- "Our work provides theoretical underpinnings of curriculum design for teaching via demonstrations, which can be beneficial for education applications and faster training of IRL/BC learners across application domains." This is a huge overstatement.

**Time Spent Reviewing:**

4

---

> ### Author Response · Authors · 2021-08-10
> **Response to Reviewer kSL2**
>
> Thank you for carefully reviewing our paper! We greatly appreciate your feedback. Please see below our responses to your comments. We have also clarified a misunderstanding regarding the scale of the environments in our experimental evaluation. We hope that our responses can address your concerns and are helpful for improving your rating. If you have any other comments or feedback, please let us know! We are looking forward to hearing back from you!
>
> -----
> **1. The review of literature was somewhat limited. In particular, there are several papers by Mark Ho and Michael Littman and colleagues that seem relevant.**
> - Thank you for the feedback. As suggested, we will expand on the literature review in the revised paper. In particular, we will add the following: (a) papers from the researchers suggested by the reviewer as listed below, (b) papers on the topic of teaching dimension and optimal cooperative inference.
>
> [Ho et al. 2021] M. K. Ho, F. Cushman, M. L. Littman, J. L. Austerweil. “Communication in action: Planning and interpreting communicative demonstrations.” Journal of Experimental Psychology: General 2021.
>
> [Ho et al. 2019] M. K. Ho, F. Cushman, M. L. Littman, J. L. Austerweil. “People teach with rewards and punishments as communication not reinforcements.” Journal of Experimental Psychology: General 2019.
>
> [Ho et al. 2018] M. K. Ho, M. L. Littman, F. Cushman, J. L. Austerweil. “Effectively learning from pedagogical demonstrations.” Conference of the Cognitive Science Society 2018.
>
> [Yang et al. 2018] S. C.-H. Yang, Y. Yu, A. Givchi, P. Wang, W. K. Vong, P. Shafto. “Optimal cooperative inference.” AISTATS 2018.
>
> [Ho et al. 2017] M. K. Ho, M. L. Littman, J. L. Austerweil. “Teaching by intervention: Working backwards, undoing mistakes, or correcting mistakes?” Conference of the Cognitive Science Society 2017.
>
> [Ho et al. 2016] M. K. Ho, M. L. Littman, J. MacGlashan, F. Cushman, J. L. Austerweil. “Showing versus doing: Teaching by demonstration.” NeurIPS 2016.
>
> **2. Clarification regarding the scale of the environments in our experimental evaluation**
> - We want to clarify the reviewer’s misunderstanding about the scale of the environments considered in the paper. In particular, for the experiments in Section 6 and Appendix B.3, the agent is learning a multi-task neural policy for solving any navigation task provided as input -- crucially, the state space corresponds to all possible configurations of these navigation grids, and the state space size is combinatorial in terms of the number of objects. For instance, consider the environment in Section 6: here, the state space size in the MDP is over 10^10, corresponding to different ways of placing objects.
>
> **3. The paper is rather niche. The theoretical results appear to be limited to the two approaches considered, and not general to curriculum learning.**
> - In this response, we want to highlight the significance of our theoretical results in the context of recent theoretical works on curriculum design for supervised learning ([Weinshall et al. 2018; Weinshall and Amir 2018]) and for imitation learning ([Brown and Niekum 2019; Kamalaruban et al. 2019]).
> - In general, there are limited theoretical tools for analyzing the curriculum learning paradigm. In the past few years, there have been several exciting theoretical results on curriculum design in the context of supervised learning -- for instance, [Weinshall et al. 2018] and [Weinshall and Amir 2018] developed a theory of curriculum learning for linear regression and binary classification settings. Our work builds on the ideas of [Weinshall et al. 2018] (see our response to comment 4) and improves upon the existing theoretical results on curriculum design for imitation learning.
> - In contrast to [Brown and Niekum 2019], we develop an interactive teaching algorithm that is able to utilize the learner’s current policy and select demonstrations adapted to the learner’s progress. In contrast to [Kamalaruban et al. 2019], our teaching algorithm does not require knowledge of the learner’s update rule. Furthermore, we provide analysis for two popular imitation learner models: the CrossEnt-BC learner model and the MaxEnt-IRL learner model (the results in [Kamalaruban et al. 2019] didn’t consider CrossEnt-BC). While we focus on two popular imitation learner models for developing the theory, the insights from our work could also be potentially useful in analyzing curriculum strategies for more advanced imitation learner models in future works.
>
> [Weinshall et al. 2018] Weinshall et al. “Curriculum learning by transfer learning: Theory and experiments with deep networks.” ICML 2018.
>
> [Weinshall and Amir 2018] Weinshall and Amir. “Theory of curriculum learning, with convex loss functions.” arXiv 2018.
>
> [Brown and Niekum 2019] Brown and Niekum. “Machine teaching for inverse reinforcement learning: Algorithms and applications.” AAAI 2019.
>
> [Kamalaruban et al. 2019] Kamalaruban et al. “Interactive teaching algorithms for inverse reinforcement learning.” IJCAI 2019.
>
> **4. The motivation for difficulty scores was weak**
> - The high-level motivation for using difficulty scores is based on recent empirical works in supervised learning and sequential decision-making settings where different notions of difficulty are used in designing curriculums (e.g., references [22]--[27] in the paper). The theoretical motivation stems from the recent work of [Weinshall et al. 2018] that theoretically investigates how training examples with different difficulty scores affect the model convergence for linear regression setting. In particular, given an example, they defined these difficulty scores in terms of the underlying loss function w.r.t. the teacher’s or the learner’s parameters. Our work builds on this by introducing notions of difficulty scores over demonstrations in a sequential decision-making setting, and uses that to design a curriculum strategy for imitation learning. As noted by the reviewer, the difficulty scores can be possibly interpreted in terms of the trade-off between the teacher’s teaching effort and the learner’s performance gain. We will add a discussion on this in the revised paper.
>
> **5. The difficulty scores and differential effect of the difficulty scores**
> - The difficulty score of a demonstration $\xi$ w.r.t. a given policy $\pi_{\theta}$ (in our case, the teacher’s policy or the learner’s current policy) is a real-valued number given by Eq. 2. In particular, the difficulty score is inversely proportional to the likelihood of the demonstration $\xi$ under the policy $\pi_{\theta}$.
> - By “differential effect of the difficulty scores”, we mean to capture the effect of picking demonstrations with higher or lower difficulty scores on the learning progress. This is based on the derivative of the expected convergence rate of the teaching objective (Eq. 3) w.r.t the difficulty scores.
>
> **6. Feasible set in lines 142-143**
> - The difficulty values $\psi^E$ and $\psi^L$ are ‘fixed’ real numbered values. Given these values, {$\xi | \Psi^E(\xi) = \psi^E \text{ and } \Psi^L(\xi) = \psi^L$} denotes the set of all demonstrations $\xi$ for which the teacher’s difficulty score $\Psi^E(\xi)$ is equal to the value $\psi^E$, and the learner’s difficulty score $\Psi^L(\xi)$ is equal to the value $\psi^L$. We will clarify this notation in the revised paper.
>
> **7. Technical details on eta**
> - We will add the technical conditions on eta in the main text of the revised paper.
>
> **8. Theorem 1 and curriculum strategy in Eq 1.; the ordering of the exposition for Theorem 1 and Eq 1**
> - Theorem 1 suggests that choosing demonstrations with lower difficulty score w.r.t. the teacher’s policy and higher difficulty score w.r.t. the learner’s policy would lead to faster convergence. Our curriculum strategy (Eq. 1)  induces a ranking preference over demonstrations that aligns with these insights of Theorem 1. Furthermore, this particular form of combining the two difficulty scores given in Eq. 1 leads to a fast convergence rate (see Theorem 2) matching the state-of-the-art work (see comment 3).
> - Regarding the ordering of the exposition, we chose to introduce the curriculum strategy (Eq. 1) in Section 3 primarily to separate the teaching algorithms (Algorithms 2 and 3) from theoretical analysis (Theorems 1, 2, and 3). In particular, the teaching algorithms in Section 3 can be applied more generally beyond the specific settings considered for theoretical analysis in Section 4. For instance, in experiments, we apply these algorithms for a non-linear MaxEnt-IRL learner model and an environment with stochastic transition dynamics. We will update the writing to clarify and justify this exposition.
>
> **9. In Theorem 3, what does it mean for the convergence rate to be “roughly” something?**
> - We will clarify this in the revised paper and add more technical details from Appendices C and D to Section 4. More concretely, the proof relies on Taylor approximation, where we ignore the negligible effect of higher-order terms.
>
> **10. It is not ideal to put details relevant to understanding performance in the appendix**
> - As suggested, we will provide more details in the main paper. We would like to highlight that the same scheduling mechanism is utilized for all teachers.
>
> **11. Other detailed comments and unclear points**
> - We will carefully revise the paper to fix the unclear points. We will clarify the meaning of the following phrases: (a) “domain knowledge”, (b) “by probing the learner”, (c) “leads to CUR teaching algorithm”, (d) “self-curriculum”, and (e)  “when the teacher’s optimal policy is unknown”. We will provide a more detailed description and motivating context for the experiments. We will also appropriately update the conclusions.
>
> -----
> We hope that our responses can address your concerns. If you have any other comments or feedback, please let us know! We will be happy to provide further responses. Thank you again for the review!

---

> > ### Comment · Reviewer_kSL2 · 2021-09-01
> > **Response to rebuttal**
> >
> > Thanks to the authors for their detailed response. After reading through the other reviews, responses, and responses to the reviews, I continue to be lukewarm on the paper. One aspect, that perhaps could be addressed in a revision, is the wide variety of the literature that the authors acknowledge is relevant but was never reviewed in the original submission. This is a real challenge to understanding the paper, then and now. I believe the lack of clarity throughout the paper, which the authors have worked hard to resolve is a larger issue than is acknowledged. I wish the authors the best with this line of work!

---

> > > ### Author Response · Authors · 2021-09-02
> > > **Thank you for the feedback (Reviewer kSL2)**
> > >
> > > We sincerely thank the reviewer for the valuable feedback. As suggested by the reviewer, we will revise the paper appropriately and have a more detailed discussion about the related literature. Also, we will carefully incorporate the reviewer’s suggestions to improve clarity. Thank you again for the review!

---

> ### Author Response · Authors · 2021-08-26
> **Re: Response to Reviewer kSL2**
>
> We thank the reviewer again for their comments. We sincerely hope that our detailed responses have addressed your concerns and are helpful for improving your rating. If you have any other comments or feedback, please let us know! We will be happy to provide further responses. Thank you again for the review.

---

### Comment · Area_Chair_roZL · 2021-09-18
**Curriculum Design: Significance of the contribution compared to Kamalaruban et al, IJCAI 2019 ?**

Dear authors and dear reviewers,

I would like us to discuss the significance of the contribution of the present paper compare to the (cited)  related work, "Parameswaran Kamalaruban, Rati Devidze, Volkan Cevher, and Adish Singla. Interactive teaching algorithms for inverse reinforcement learning. IJCAI 2019".

Thanks in advance, Area Chair

---

> ### Author Response · Authors · 2021-09-19
> **Re.: Curriculum Design: Significance of the contribution compared to Kamalaruban et al, IJCAI 2019 ?**
>
> Dear Area Chair,
>
> Thank you for the comment. In our response below, we discuss the key technical differences between [Kamalaruban et al. 2019] and our submission. These technical differences, in turn, allow our approach to overcome several limitations of the related work.
>
> - **[Kamalaruban et al. 2019]**: The omniscient teaching algorithm of [Kamalaruban et al. 2019] is based on an iterative machine teaching framework that picks the next demonstration to steer the learner towards a target parameter. More concretely, at time $t$, the teaching algorithm picks the demonstration based on the knowledge of the teacher's target parameter $\theta^*$, the learner's internal parameter $\theta_t$, and the learner's internal update rule in the form of the gradient $g_t$. However, assuming access to both the teacher's target parameter $\theta^*$ and gradients $g_t$ limits the applicability of their omniscient teaching algorithm in important motivating scenarios: (a) in teacher-centric applications (e.g., tutoring systems), the target teaching parameter $\theta^*$ could be known, however, computing the gradient $g_t$ requires access to the specific learner model; (b) in learner-centric applications (e.g., self-curriculum by an agent), the gradient $g_t$ could be computed, however, the target teaching parameter $\theta^*$ is unknown. We note that their paper also proposed a heuristic blackbox teaching algorithm that uses an empirical estimate of $\theta_t$ and replaces the per-step optimization problem of the omniscient algorithm with an approximated functional form. However, in comparison to the omniscient algorithm, this heuristic blackbox algorithm does not have any theoretical guarantees. Furthermore, the approximated functional form used in their blackbox algorithm is specifically based on the MaxEnt-IRL / MCE-IRL learner's internal update rule.
>
> - **Our submission**: The curriculum strategy proposed in our submission takes a completely different perspective -- it picks the next demonstration based on a preference ranking induced by the teacher's and learner's policies. More concretely, at time $t$, the ranking is obtained based on the ratio between a demonstration's difficulty scores under the learner's current policy $\pi_t$ and the teacher's policy $\pi^*$. Crucially, in comparison to the teaching algorithms of [Kamalaruban et al. 2019], our curriculum strategy does not assume the knowledge of the learner's internal update rule (i.e., the form of the gradient $g_t$). Also, note that our curriculum strategy only requires access to the learner's current policy $\pi_t$ and not the learner's internal parameter $\theta_t$ (this is useful as the learner might have a different representation structure). Furthermore, in our response to Reviewer RaMd, we have provided additional results on how our curriculum strategy can be applied in a setting with limited observability of $\pi_t$.
>
> Below, we discuss the benefits and significant contributions of our approach compared to the related work of [Kamalaruban et al. 2019].
>
> - **Unified curriculum strategy (Section 3)**: We present a unified curriculum strategy through a preference ranking over demonstrations that is more widely applicable for different learner models and application scenarios (i.e., both the teacher-centric and learner-centric settings). This ranking is obtained based on the ratio between a demonstration's likelihood under the teacher's policy and the learner's current policy. Notably, one does not require knowledge of the learner's update rule when implementing our curriculum strategy. In contrast, the teaching algorithms of [Kamalaruban et al. 2019] are specifically designed for the MaxEnt-IRL / MCE-IRL learner model.
>
> - **Theoretical analysis (Section 4)**: For our analysis, we consider MaxEnt-IRL and CrossEnt-BC learner models, and theoretically justify the choice of preference ranking over demonstrations (Theorems 1 and 3). In Theorem 2, we show that our curriculum strategy achieves a linear convergence rate for the MaxEnt-IRL learner model under the setting of deterministic transitions and linear reward function. Importantly, this rate is achieved by our curriculum strategy that does not rely on specifics of the learner model when selecting demonstrations, and matches the rate by the omniscient teaching algorithm from [Kamalaruban et al. 2019] that requires full knowledge of the learner model.
>
> - **Experiments in the teacher-centric setting (Section 5)**: In our first set of experiments, we consider the teacher-centric setting, and show the effectiveness of our proposed curriculum strategy by considering different learner models (MaxEnt-IRL and CrossEnt-BC). For both the learner models, we show that our curriculum strategy is competitive w.r.t. the natural baselines. On the other hand, the omniscient teaching algorithm of [Kamalaruban et al. 2019] is not applicable for the CrossEnt-BC learner model.
>
> - **Experiments in the learner-centric setting (Section 6)**: In our second set of experiments, we consider the learner-centric setting where an agent uses our curriculum strategy for designing a self-curriculum over a given set of training demonstrations. Here, we consider a more challenging environment where the agent has to learn a multi-task neural policy for solving navigation tasks. The experimental results show the utility of our curriculum strategy in comparison to the i.i.d. sampling over training demonstrations. In contrast, the work of [Kamalaruban et al. 2019] is only applicable to the teacher-centric setting, and cannot be directly extended to the learner-centric setting.
>
> We hope that the above response highlights the significance of the contributions of our submission. If you have any other comments or feedback, please let us know!
>
> Thank you,
>
> Authors

---

> > ### Comment · Reviewer_RaMd · 2021-09-22
> > **Overstating novelty**
> >
> > Thank you for your response. I agree that the submission contains novel results; however, after reading Kamalaruban et al. 2019, it appears that the claims of novelty as stated in the current submission are significantly exaggerated. As the authors now acknowledge, their paper is not the first to consider the black box learning setting where the teacher does not have access to the learner's internal parameters or update rule. I think the paper should have been more open about this prior work and would have been stronger as a result of a more careful juxtaposition between the current submission and prior blackbox teaching. Prior work also only requires access to the learners policy (and seems to only requires sample access, whereas the current approach seems to require the entire policy).  Thus, Kamualaruban et al. seems like an important (but missing) baseline to compare against the current approach.

---

> > ### Author Response · Authors · 2021-09-22
> > **Re.: Curriculum Design: Significance of the contribution compared to Kamalaruban et al, IJCAI 2019 ?**
> >
> > We would like to highlight again that the approximated functional form used in their blackbox algorithm is specifically based on the MaxEnt-IRL / MCE-IRL learner's internal update rule. It becomes more evident when the teacher’s reward function is linear given by $R^E(s, a) = <w^*, \phi(s,a)>$. Then, the blackbox algorithm's per-step optimization problem is simply $|<g_t, w^*>|$ where $g_t$ is the gradient of the linear MCE-IRL learner model. Note that this per-step optimization problem can be written in the form $|\sum_{s,a} \big(\rho^{\pi_t}(s,a) - \rho^{\xi}(s,a)\big) R^E(s, a)|$ as used in their algorithm. As the Reviewer RaMd has suggested, this blackbox algorithm could also be used with a CrossEnt-BC learner model -- however, this has not been validated in their work and has no principled justification.
> >
> > The main reason for only adding the omniscient teaching algorithm of [Kamalaruban et al. 2019] as a baseline is because we did all experiments under full observability of the learner's policy.  As the Reviewer RaMd has suggested, we will also add their blackbox teaching algorithm as a baseline in the revised paper.

---

> > > ### Comment · Reviewer_RaMd · 2021-09-22
> > > **bbox algorithm clarification**
> > >
> > > Thank you for the clarification. I see your point that the prior blackbox approach is specifically tailored to the MCE-IRL objective. I think this is an important clarification that does improve the novelty of the current submission. I think this deserves more discussion in the submission, but I see now how the authors' claim of being the first to study blackbox teaching without knowing the learner's dynamics can be justified. Kamalaruban et al. claim that their blackbox teaching approach does not know the learner's dynamics which is also the same claim made in the current submission. I think I agree with the authors that the strategy of Kamalaruban et al. is still closely tied to the MCE-IRL algorithm so it is unclear if they can accurately claim to be agnostic about the dynamics. Is that an accurate interpretation?
> > >
> > > I agree that the current approach is agnostic to the learners dynamics and I think this is a nuanced, but significant improvement over prior work that merits publication.

---

> > > ### Author Response · Authors · 2021-09-23
> > > **Re. bbox algorithm clarification**
> > >
> > > We sincerely appreciate the Reviewer RaMd's engagement and follow-up comments.
> > >
> > > Yes, we agree with the reviewer's interpretation. Basically, one could always design a heuristic learner-agnostic algorithm by simply assuming a specific learner model for building the algorithm and then applying it to other learner models. In this sense, the blackbox algorithm of [Kamalaruban et al. 2019] is built by considering the gradient functional form of the linear MCE-IRL learner model.
> > >
> > > We again want to highlight the significant differences between [Kamalaruban et al. 2019] and our work. [Kamalaruban et al. 2019] develop the omniscient algorithm using an iterative machine teaching framework -- this algorithm is central to their work for which they have done extensive theoretical analysis. Then, they develop a heuristic blackbox algorithm by simply replacing the objective of the omniscient teacher and considering the gradient functional form of the linear MCE-IRL learner model. However, these algorithms from [Kamalaruban et al. 2019] are very much tied to MCE-IRL learner models and are teacher-centric (e.g., their blackbox algorithm still needs access to the true reward function $R^E$). The above reasoning also explains our motivation for only using their omniscient algorithm as a baseline in our experiments.
> > >
> > > In contrast, our proposed curriculum strategy takes a completely different perspective to develop an algorithm that is agnostic to the learner's dynamics. It is important to note that the difficulty scores, central to our algorithm, are much more applicable in real-world settings. For instance, even in the teacher-centric setting, it is quite plausible that the teacher has access only to an expert policy $\pi^E$ but not a well-specified function $R^E$. In summary, we sincerely believe that our work brings significant contributions compared to [Kamalaruban et al. 2019] by developing principled curriculum strategies that are agnostic to the learner's dynamics and are applicable in both the teacher-centric as well as learner-centric settings.

---

> > > > ### Comment · Reviewer_RaMd · 2021-09-23
> > > > **thank you**
> > > >
> > > > Thank you for the clarifications. I think this issue is more nuanced than originally suggested in the submission; however, I am convinced now that the paper is indeed novel in the ways originally claimed and should be accepted.

---

### Decision · Program_Chairs · 2021-09-27

**Decision:**

Accept (Poster)

**Comment:**

The paper is interested in defining an interactive curriculum for learning from demonstrations. Like in Weinshall et al., 2018, the idea is that the most useful demonstrations are the easiest ones for the teacher/oracle and the most difficult ones for the current learner (where the difficulty of a demonstration w.r.t. a policy is computed as its probability w.r.t. this policy).
The lesson is backed by a theoretical analysis in the linear case, considering MaxEnt-IRL and CrossEnt-BC algorithms.

Three approaches are experimentally compared: CUR (selecting the demonstration less difficult wrt the expert and most difficult wrt the current learner), CUR-L (where the difficulty wrt the expert is replaced with a heuristic measure) and CUR-T (that mostly considers the difficulty wrt the expert). The results suggest that the most important is to take into account the difficulty w.r.t. the current learner.

The area chair did not take into account the most negative review (nb: this reviewer was eventually convinced by the good and thorough job done by the authors w.r.t. all reviewers, to explain the issues and to promise to revise their paper w.r.t. all points and suggestions raised by the reviewers).

However, the area chair considered that the novelty of the paper w.r.t.  Kamalaruban et al.'s paper  (IJCAI 2019, cited) needed to be discussed:

* a main difference lies in  considering MaxEnt-IRL and CrossEnt-BC whereas Kamalaruban et al. considered MCE-IRL;

* the structure of the proof (section 4.3 in Kamalaruban et al; Appendix C in the submitted paper) is similar ;

* the experimental setting (the car driving problem) is the same;

* this paper argues that it improves on Kamalaruban et al as it does not require the knowledge of the optimal policy, as it replaces the difficulty w.r.t. the expert policy with a heuristic function. The quality of this heuristic function however does not seem to be very important (point 3 in Response to Reviewer RaMd); unless I missed it, the authors did not comment further on it and did not consider any lesion of this, to study its impact.

The significance of the contributions in the submitted paper w.r.t. Kamalaruban et al's (K, in the following) has been extensively discussed, concluding that
* K presents an omniscient algorithm, aimed to yield the optimal $\theta^*$ policy parameter using the gradient $g_t$
* K also presents an approximation thereof, based on an estimation of $\theta_t$, though with no guarantees and specifically based on the MaxEnt-IRL / MCE-IRL freamework.


Overall, the paper is considering a hot topic and presenting interesting results; the many additional complements (explanations, discussion and results) in the answers to the reviews let us hope that the revised version will be way better than the submitted one.
It is mandatory that in the final version, the authors clarify the novelty issue, and  compare their results with the Blackbox teaching baseline (section 5 in K).